# An in-solution snapshot of SARS-COV-2 main protease maturation process and inhibition

Gabriela Dias Noske[1,6], Yun Song [ORCID][2,6], Rafaela Sachetto Fernandes[1], Rod Chalk[3], Haitem Elmassoudi[3], Lizbé Koekemoer[3], C. David Owen[2], Tarick J. El-Baba[4,5], Carol V. Robinson [ORCID][4,5], The COVID Moonshot Consortium*, Glaucius Oliva [ORCID][1] & Andre Schutzer Godoy[1] ✉

The main protease from SARS-CoV-2 (M^pro) is responsible for cleavage of the viral polyprotein. M^pro self-processing is called maturation, and it is crucial for enzyme dimerization and activity. Here we use C145S M^pro to study the structure and dynamics of N-terminal cleavage in solution. Native mass spectroscopy analysis shows that mixed oligomeric states are composed of cleaved and uncleaved particles, indicating that N-terminal processing is not critical for dimerization. A 3.5 Å cryo-EM structure provides details of M^pro N-terminal cleavage outside the constrains of crystal environment. We show that different classes of inhibitors shift the balance between oligomeric states. While non-covalent inhibitor MAT-POS-e194df51-1 prevents dimerization, the covalent inhibitor nirmatrelvir induces the conversion of monomers into dimers, even with intact N-termini. Our data indicates that the M^pro dimerization is triggered by induced fit due to covalent linkage during substrate processing rather than the N-terminal processing.

Severe acute respiratory syndrome coronavirus 2 (SARS-CoV-2) is the causative agent of COVID-19[1]. Like SARS-CoV and Middle East Respiratory Syndrome coronavirus (MERS-CoV), SARS-CoV-2 is single-strand RNA virus (ssRNA) that belongs to the genera of the beta-coronaviruses[1,2]. The SARS-CoV-2 genome is composed of nearly 30,000 nucleotides which contains the *ORF1ab* gene, a large open-reading-frame (ORF) responsible for encoding 16 non-structural proteins (nsp's) as two polyproteins after ribosomal frameshifting, named 1a and 1b[1–3]. Proteolytic processing of viral polyproteins is essential for the viral life cycle, and it's performed by two SARS-CoV-2 encoded cysteine proteases: the papain-like protease (PL^pro), which is one of the domains of nsp3, and the viral main protease (M^pro or 3CL^pro), encoded by the nsp5[4,5]. PL^pro cleaves the viral polyprotein at three sites, while M^pro is responsible for cleaving the polyprotein at eleven distinct sites[4,5], including its own N- and C-termini[6]. M^pro is one of the most

promising targets for drug development against SARS-CoV-2[5,7]. All the gathered structural and biochemical information on this target has been crucial for the rapid development of new antivirals[8–11], including the recent drug Paxlovid/nirmatrelvir[10], approved for emergency use in USA, Europe and China.

To obtain heterologous expressed mature M^pro, researchers adopted the general strategy of adding nsp4 C-terminal portion to the N-terminal portion of nsp5 constructs, allowing the self-cleavage and dimerization of M^pro in-vitro[6]. Previously, we described the activity and biochemical profile of SARS-CoV-2 M^pro C145S mutant containing the C-terminal portion of nsp4 at its N-termini[6]. This serine mutation generated an active but much slower version of M^pro, a valuable tool for studying the biochemical aspects of this enzyme. Differently than the dimeric M^pro, this sample behaved as a dynamic mix of monomers, dimers, trimers and tetramers in solution[6]. The residual activity of this

[1]Sao Carlos Institute of Physics, University of Sao Paulo, Av. Joao Dagnone, 1100 - Jardim Santa Angelina, Sao Carlos 13563-120, Brazil. [2]Electron Bio-imaging Centre, Diamond Light Source Ltd., Harwell Science and Innovation Campus, Didcot OX11 0QX, UK. [3]Centre for Medicines Discovery, Oxford University, OX1 3QU Oxford, UK. [4]Physical and Theoretical Chemistry Laboratory, Department of Chemistry, University of Oxford, South Parks Road, OX1 3TA Oxford, UK. [5]The Kavli Institute for Nanoscience Discovery, Dorothy Crowfoot Hodgkin Building, South Parks Road, OX1 3QU Oxford, UK. [6]These authors contributed equally: Gabriela Dias Noske, Yun Song. *A list of authors and their affiliations appears at the end of the paper. ✉e-mail: andregodoy@ifsc.usp.br

serine mutant permitted the slow cleavage of N-termini nsp4-nsp5 peptide, which allowed us to monitor the oligomeric states of M$^{pro}$ in solution during the maturation process[6]. We observed that the tetrameric form of M$^{pro}$ C145S achieves the equilibrium as dimers over the course of two days, in a phenomenon that is directly proportional to the N-termini cleavage, therefore inferring that the N-terminal processing was critical for dimerization[6]. Additionally, we previously revealed the crystal structure of M$^{pro}$ C145S in complex with the nsp5-nsp6 C-terminal peptide, obtained from the tetrameric M$^{pro}$ sample, presenting a detailed depiction of this key cleavage maturation event. In summary, two mature M$^{pro}$ dimers assemble in a dimer-dimer transient complex, positioning the C-terminal of one M$^{pro}$ molecule towards the active site of another, and followed by C-terminal processing in a trans-cleavage event[6].

Here we report the cryo-electron microscopy (cryo-EM) structure of SARS-CoV-2 M$^{pro}$ C145S mutant containing the nsp4-nsp5 cleavage portion, at 3.5 Å resolution. In this structure, we glimpse the in-solution state of M$^{pro}$, bound to two unfolded nsp5 protomers during N-terminal processing. We demonstrate that dimerization is not dependent of N-terminal processing, as previously assumed, but likely a consequence of induced fit required for the covalent linkage during the substrate processing.

## Results

### Native mass analysis of SARS-CoV-2 M$^{pro}$ C145S

Native mass spectroscopy analysis of peaks containing multimers of SARS-CoV-2 M$^{pro}$ C145S revealed that the sample is composed of a mix of monomers, dimers, trimers and tetramers (Fig. 1), as previously indicated by SEC-MALS analysis[6]. Additionally, we observed monomer particles that contain sequences of folded and unfolded uncleaved nsp4-nsp5 peptide. In line with our previous model of maturation, in which the cleavage of N-terminal peptide serves as the trigger for dimerization.

However, native mass spectroscopy also revealed that the peaks containing oligomeric states could be formed by the combination of uncleaved (theoretical mass of 34,554.54 Da) and cleaved (theoretical mass of 33,780.58 Da) nsp4-nsp5 particles. The same goes both for trimers or tetramers peaks, that surprisingly could be composed by all combinatory possibilities of cleaved and non-cleaved peptides (Fig. 1 and Supplementary Fig. 1). This contradicts preliminary models of M$^{pro}$ maturation, in which N-terminal processing dictates the dimerization[6,12,13]. It is also clear by the mass relative quantities that the equilibria of oligomeric states favor the states where more cleaved elements are present (Fig. 1 and Supplementary Fig. 1), still befitting with that model were N-terminal cleavage is directly involved in dimerization.

To confirm these results, we performed a secondary native mass spectroscopy analysis of sample 2 using different setup. Using an Orbitrap Q-Exactive UHMR, seven charge state distributions were observed. We compared the data of with the previous run from Agilent instrument in Supplementary Fig. 6. An abundant charge state distribution near m/z - 2500 and centered at 11+ ion corresponds to M$^{pro}$ with molecular mass $34,556 \pm 24$ Da (m1 in Supplementary Fig. 6). A low-abundance charge state distribution with a similar magnitude of the charge is also observed, which corresponds to a species with mass $33,865 \pm 44$ Da (m2 in Supplementary Fig. 6). These two charge state distributions can be assigned to the different M$^{pro}$ monomers. Multiple peak series can be found between m/z 4000–5000 (all of which are centered at charge state 16 +), which correspond to ions with molecular weights consistent with m1m1 homo- and m1m2 heterodimers, i.e., $67,687 \pm 44$ Da for homodimers and $68,466 \pm 9$ Da for heterodimers. Interestingly, the additional peaks in the spectrum correspond

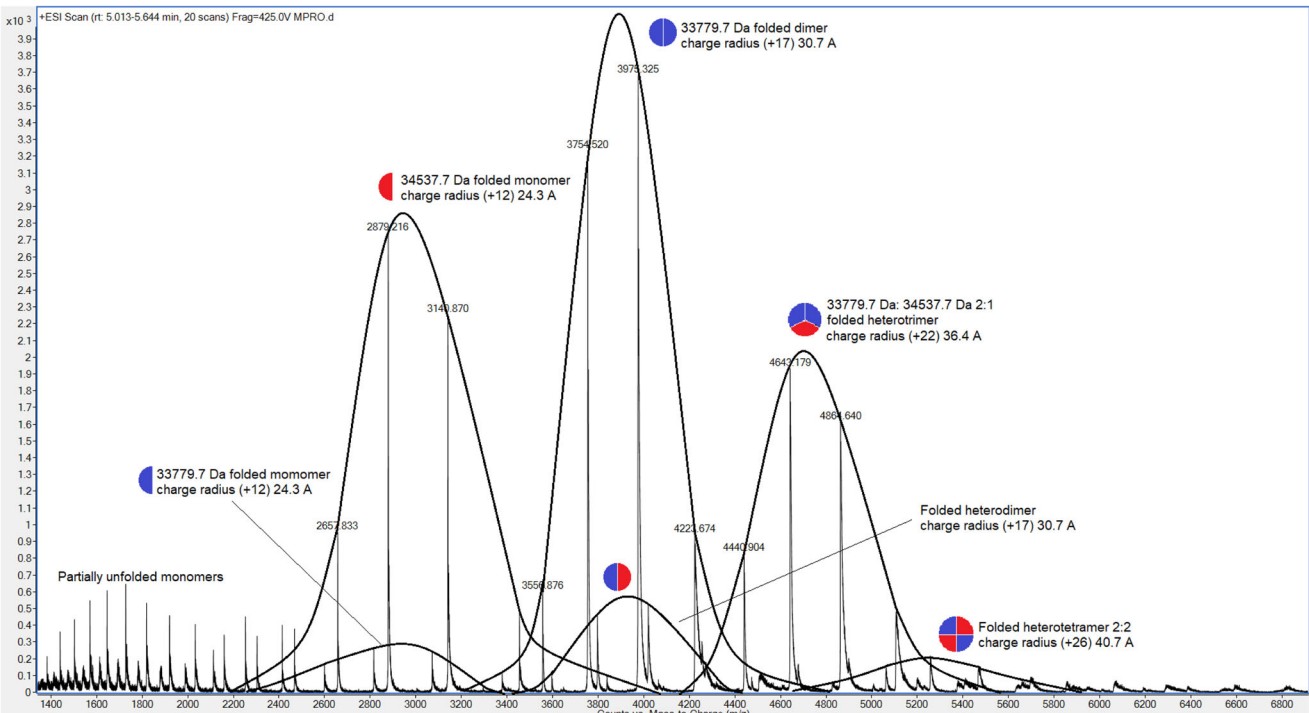

**Fig. 1 | Native mass spectrometry of C145S SARS-CoV-2 M$^{pro}$.** From left to right, peaks show monomers cleaved (blue semicircle) an uncleaved (red semicircle), dimers formed by cleaved (blue circles) or half-cleaved (blue-red circle) particles, trimers formed by two cleaved and one uncleaved particles (two thirds blue, one-third red circles) and tetramers formed by two cleaved and two uncleaved particles (two-quarters blue, two-quarters red). Minor peaks of other forms are described in supplementary materials. Graphs were plotted from individual native mass spectrometry experiments.

to charge state that deconvolute to molecular weights that are consistent with higher order oligomeric states, with different amounts of m1 and m2. Still, the data shows that sample 2 is composed of native particles combining cleaved and uncleaved protomers, confirming our previous observation.

## Cryo-EM reveals in solution details of N-terminal processing

Here we describe the cryo-Em structure of SARS-CoV-2 M[pro] C145S at 3.5 Å (Fig. 2). The final model showed all 306 residues from C145S M[pro] visible for both chains, plus clear evidence of an at least 11 residues long peptide occupying both active sites (Fig. 3a). The final model showed no rotamers, Cβ or Ramachandran outliers, and a map-model correlation coefficient (CC) of 0.80 (Fig. 3b, c). Model and/or map were deposited under the codes 8EY2 (for PDB) and EMD-28666 (for EMDB). Final cryo-EM model is very similar to X-ray known structures (Fig. 3d), with an RMSD of 0.7 Å (for 3,909 atoms) when compared with the mature dimeric form of M[pro] (PDB 7KPH), or 1.2 Å (for 3,822 atoms) when compared with X-ray structure of C145S mutant (PDB 7N5Z) (Fig. 3d and e). Statistics and parameters from data collection and processing are available in Table 1.

It was previously assumed that the tetrameric sample of M[pro] C145S observed in the solution had the same organization than the one obtained with crystal structure of dimer-dimer association[6]. However, the cryo-EM structure of M[pro] C145S revealed a dimeric particle of M[pro] anchored to the nsp4-nsp5 region in the moment that precedes cleavage, with detailed density of P and P' key residues (Fig. 3a). The 3.5 Å resolution structure provides structural insights into an important step of M[pro] maturation, the N-terminal cleavage, exhibiting a clear electron density of the peptide in both chains A and B of the dimer (Fig. 3a). The electric potential map around the active site Ser145* indicates a non-covalent interaction of the nsp4/nps5 peptide (residues SAVLQ of nsp4 and residues 1-6 of nsp5 M[pro], SGFRKM), forming an extended β-sheet maintained mostly by hydrogen bonds at Thr24, Gly143, Ser145*, His163, Glu166 and Gln189 (Fig. 4a). Substrate positions exhibited a binding position highly like the crystallographic structures of M[pro] in complex with the physiological peptide substrate (PDBid 7N6N and 7DVP) with RMSD values of respectively of 0.57 and 0.9 calculated between 30 and 54 atoms.

In the cryo-EM model, we can see Gln0 occupying the S1 subsite, with its NE2 side chain moiety interacting with OE1 atom of residue Glu166 through a hydrogen bond (2.7 Å and 2.5 Å for chains A and B, respectively). Additionally, OG atom of Ser145* forms a hydrogen bond with main chain hydroxyl group of Gln0. The Gly143 main chain amide donates a hydrogen bond to P1' carbonyl oxygen, stabilizing the oxyanion hole during catalysis. Leu-1 at the S2 subsite interacts with Gln189 through a hydrogen bond formed between Leu-1 main chain amide and Gln189 side chain OE1 (Fig. 4a, b). Met49, Met149 and Gln189 from the S2 subsite assume a more open conformation compared with captured unbound forms of the enzyme. At the S3 subsite, there is an interaction between the polar atoms of the Val-2 main chain and Glu166 side chain (Fig. 4c). The hydroxyl group of main chain Ala-3 in P4 donates a hydrogen bond to NE2 of Gln189. The lack of polar interactions between protein sites S2-S4 with protein residues would help to explain the variety of distinct amino acids these positions are willing to accommodate. The density map of active site anchored M[pro] particle only extends towards residues Met6 from M[pro], suggesting that the rest of the particle is too mobile for model reconstruction.

By carefully inspecting low pass filtered particles used for final reconstruction, we noticed an elongated satellite particle anchored around each one of the dimer active sites regions, with size and shape consistent with a partially folded monomer of M[pro] (Fig. 4d). Contrasting with the dimer-dimer complex formed during the C-terminal cleavage, these elongated protomers are randomly distributed around the active site, indicating that the particles are misfolded (Fig. 4d, e). Moreover, the elongated particles shaped as nsp5 monomers does not

seem to form dimers (Fig. 4e). These observations are in line with our previous model of immature M[pro], in which we showed that the addition of three non-cleavable amino acids to its N-termini disrupts M[pro] folding and prevents dimerization[6]. Nevertheless, our structure reveals a unique view of the dimeric form of M[pro] caught during the cleavage of the N-termini in solution. We could not generate 2D or 3D classes highlighting the satellite particles, which agrees with the partially folded hypothesis.

## 3D variability analysis reveals in solution flexibility of the active site

Many in-silico molecular simulations and docking studies have tried to describe M[pro] in solution movement and substrate recognition[14,15], no experimentally determined model is available to validate these free of the enclosure of crystal constrains. In here, we used the cryo-EM model to study the dynamic of M[pro] during N-terminal cleavage. For that, the particles used for model reconstruction were analyzed using cryoSPARC 3DVA tool for sorting in solution variability of M[pro]. The first of four eigenvector components explored generated low-quality volumes that were not interpretable. The other three (components 2, 3 and 4) produced volume frames that were used for analysis of particle variability. In all three, one can see a volume with shape of M[pro] bound to the nsp4 peptide in both active sites, suggesting that particles of other oligomeric states were filtered during data processing. As expected, the analysis indicate that the highest conformational plasticity is concentrated in the active site region of M[pro], with the highest RMSD concentrated in the regions of helix 43-53, and in the β-hairpins 19-27, 62-65 and 166-171. Helix 43-53 contains key residues that are involved in the shaping of hydrophobic subsite S2, including Met49. Together with Met165, Met49 form a substrate recognition cavity, with a preference for hydrophobic side chains such as leucine and valine[6]. The β-hairpin 19-27, responsible for the formation of S1' subsite also appears to be remarkably mobile when compared with the rest of the protein.

The analysis indicates that both active sites are gradually expanding and contracting over the frames (Supplementary Fig. 2). The expansion of M[pro] active site was already correlated with the substrate and ligand binding[6,16], so it is likely that this variability is correlated with distinct moments of substrate anchoring. The conformational flexibility of M[pro] active site during substrate and ligand binding has been previously observed using cryogenic and room-temperature X-ray crystallography[6,17]. Kneller et al. demonstrated that during ligand binding, secondary structural elements that form P2-P5 subsites are driven from original position for ligand/substrate accommodation up to 2.4 Å, shifting the shape and electrostatic potential of the site[17]. The expansion-contraction movements of the active site seem to be symmetrical for both M[pro] protomers (Supplementary Fig. 2). This diverge significantly from multiple in-silico studies that have demonstrated that M[pro] protomers are independently active[18–20], but would help to explain why dimerization is critical for full enzyme efficiency. The active site plasticity of in-solution coronaviruses M[pro] is critical to understanding the enzyme dynamics and should assist drug discovery campaigns, especially for those platforms relying on in-silico analysis of protein-ligand interaction, which often do not correlate with experimental data[21].

## MAT-POS-e194df51-1 cause accumulation of unfolded particles

The C145S M[pro] sample was shown to be a valuable tool to investigate the in-vitro behavior of M[pro] [6]. Here we explore the capacity of our method to explore the study the effect of potent inhibitors on the enzyme maturation. Our first object was the competitive non-covalent inhibitor MAT-POS-e194df51-1, developed by the COVID Moonshot initiative, with a pIC$_{50}$ of 7.5. Samples containing monomeric C145S M[pro] (sample 1) or tetrameric C145S M[pro] (sample 2) were incubated with two concentrations of MAT-POS-e194df51-1, and their oligomeric state was monitored over the course of 48 h by SEC-MALS and

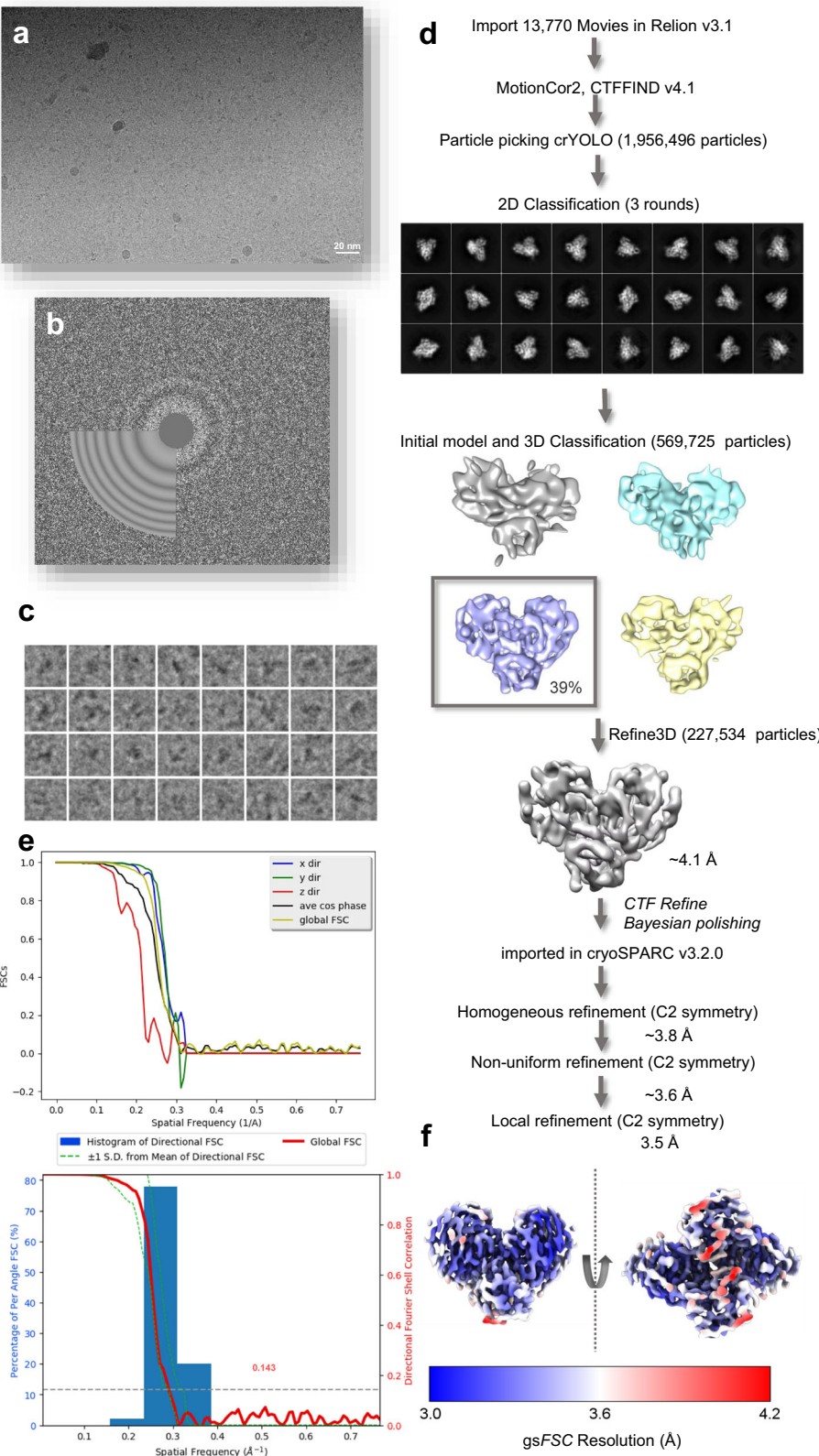

**Fig. 2 | Cryo-EM data processing schematic for C145S SARS-CoV-2 M^pro. a** Aligned micrographs, with scale bar at the bottom. **b** CTF-function calculated from obtained micrographs. **c** Extracted particles examples. **d** Detailed schematic of steps taken for final reconstruction, highlighting obtained 2D and 3D classes, and first high-resolution reconstruction. **e** Fourier shell correlation (FSC) between half maps of the final reconstructions. At the top, graph shows FSCs versus spatial frequency calculated in directions x (blue), y (green) and z (red). Average cos phase is in black, and global FSC is plotted in yellow. At the bottom, percentage of per angle FSC (blue) overlaid with gold standard FSC plot (red). **f** Local resolution projected on the final map from two orientations.

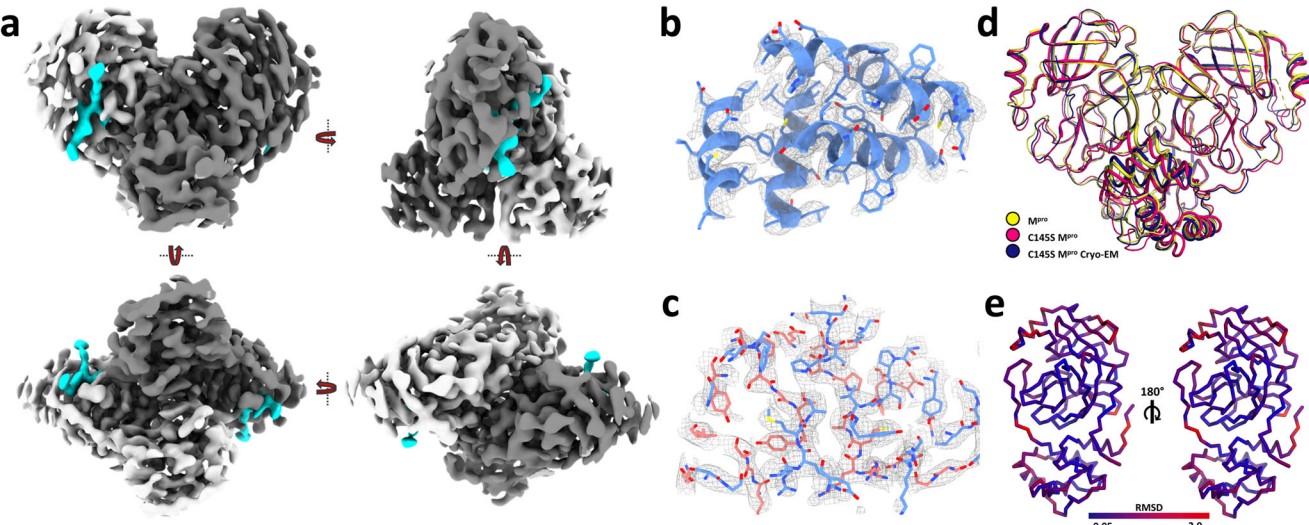

**Fig. 3 | A. Overview of SARS-CoV-2 C145S M^pro cryo-EM model. a** Four-sides rotation view of final map displayed as surface, with chains A and B coloured in white and grey, respectively, and active site peptide map coloured in cyan. **b** Chain A (blue) domain III model fitted into final map (grey). **c** Chain A (blue) and B (salmon) interface region fitted into final cryo-EM map (grey). **d** Superposition of X-ray M^pro model (yellow, PDB 7KPH), X-ray SARS-CoV-2 C145S M^pro (pink, PDB 7N5Z) and SARS-CoV-2 C145S M^pro cryo-EM model (dark blue). **e** SARS-CoV-2 C145S M^pro cryo-EM model chain A (left) and (right) coloured according its RMSD versus X-ray model of M^pro (PDB 7KPH).

compared with control. For simplification, trimers were considered to be part of the tetramers pool.

At 0 h, sample 1 control presented a 1.0/0.0 (monomer/dimer) ratio, progressing to 0.88/0.12 at 24 h and 0.39/0.61 at 48 h (Fig. 5a). Regarding sample 1 containing 4:1 MAT-POS-e194df51-1/protein, it remained at a ratio of 1.0/0.0 monomer/dimer between 0 h and 24 h, only progressing to 0.004/0.996 by the 48 h timepoint (Fig. 5b). Reducing the MAT-POS-e194df51-1/protein molar ratio from 4:1 to 0.4:1 returned the behavior to closer to that of the control, reaching a monomer/dimer ratio of 0.89/0.11 at 24 h and 0.74/0.25 at 48 h. Sample 1 showed that unfolded monomers are basically forming fully mature dimers over time. MAT-POS-e194df51-1 to sample 1, both concentrations seem to inhibit the dimer maturation and cause the accumulation of unfolded monomers particles. With this experiment we were able to demonstrate a unique effect of MAT-POS-e194df51-1 on the enzyme cycle that was never saw, showing not only its capacity of competing with the substrate but also blocking the enzyme maturation cycle and accumulate uncleaved polyprotein. In 2018 Constant et a.l demonstrated for Dengue protease NS3 that the most effective inhibitors should specifically target cleavage sites that would accumulate uncleaved viral protein precursors[22]. In this case, this inhibited phenotype might *trans*-dominantly inhibit other in-cell viral phenotypes, possibly suppressing the generation of drug-resistant variants. More studies will be necessary to comprehend the effect of non-cleaved elements in the SARS-CoV-2 metabolism.

For control of sample 2, we saw the ratio of 0.43/0.34/0.23 monomer/dimer/tetramer protein mass detected at 0 h, changing to 0.09/0.89/0.02 at 24 h and 0.0/0.98/0.02 at 48 h (Fig. 6a). For sample 2 containing 4:1 MAT-POS-e194df51-1/protein, we saw the ratio of 0.49/0.45/0.05 monomer/dimer/tetramer protein mass detected at 0 h, changing to 0.49/0.51/0.0 at 24 h and 0.45/0.55/0.02 at 48 h (Fig. 6b). For sample 2 containing 0.4:1 MAT-POS-e194df51-1/protein, we saw the ratio of 0.46/0.47/0.06 monomer/dimer/tetramer protein mass detected at 0 h, changing to 0.26/0.73/0.0 at 24 h and 0.02/0.96/0.02 at 48 h.

On the control of this experiment, we saw that the sample initiates as a mix of tetramers, dimers and monomers. Over the course of 48 h, both monomers and tetramers are extinguished, and dimers seen to became predominant. The presence of MAT-POS-e194df51-1 on sample

2 seem to cause the same effect that in sample 1 of blocking monomer consumption, but also seem to accelerate the consumption of tetramers. If we think about the tetramer samples as being the enzyme-substrate complex, it is logical that a competitive inhibitor would have this effect on the sample as it causes the dislocation of the unfolded substrate that is bound to M^pro active site. In parallel with sample 1, sample 2 monomers conversion to dimers was also blocked by the presence of MAT-POS-e194df51-1, revealing that the inhibitor not only blocks activity but also might prevent the maturation process, possibly enhancing the antiviral effect beyond its potency to the target.

**Covalent inhibitor Nirmatrelvir induces dimerization**

The effect of covalent M^pro inhibitor Nirmatrelvir/PF-07321332 with a $pIC_{50}$ of 7.7[10] was also tested with against C145S samples. The effect of this molecule was remarkably different from that with MAT-POS-e194df51-1 (Fig. 5c). For sample 1 containing 4:1 Nirmatrelvir/protein, we saw the ratio of 0.65/0.35 between monomers/dimers protein mass detected at 0 h, changing to 0.06/0.94 at 24 h and 0.02/0.98 at 48 h (Fig. 5c). For sample 2 containing 4:1 Nirmatrelvir/protein, we saw the ratio of 0.31/0.56/0.12 between monomers/dimers/tetramers protein mass detected at 0 h, changing to 0.0/0.88/0.12 at 24 h and 0.0/0.98/0.02 at 48 h (Fig. 6c).

In sample 1, the presence of Nirmatrelvir seen to strongly induce the formation of dimers from monomers even at zero hours and enhance the ratio of dimer formation over time significantly, with almost full conversion after 24 h (Fig. 5c). In sample 2, both monomers and tetramers equilibrium also seem to be dislocated to favor dimer formation (Fig. 6c). While for the tetramer increased consumption could be explained simple by the competition between Nirmatrelvir and the substrate-enzyme complex, the accelerated ratio of monomers conversion into dimers was completely unexpected. Yet, these unique results might shed light in the details of the first step of protein maturation, the N-terminal processing.

**Covalent linkage induces M^pro dimerization**

Our initial model for this step was that the N-terminal cleavage is a mix of cis and trans-events, and its proper cleavage would eliminate the steric hindrance that would be preventing dimerization[6], which was in line with previous prosed models for SARS-CoV M^pro [12,13]. However, the

**Table 1 | Cryo-EM data collection, refinement and validation parameters and statistics**

| Data collection | |
|---|---|
| Voltage | 300 |
| Magnification | 130,000 |
| Camera | K3 super resolution |
| Slit width (eV) | 20 |
| Super-resolution pixel size (Å) | 0.3265 |
| Binned pixel size (Å) | 0.653 |
| Dose rate (e/px/s) | 18.5 |
| exposure time (s) | 1.0 |
| total dose (e/Å2) | 43.385 |
| fractions | 43 |
| dose per frame (e/Å2/frame) | 1.009 |
| defocus range (μm) | -0.5 to -2.5 |
| Movies collected | 13,770 |
| **Data processing and refinement** | |
| Particles | 227,533 |
| Chains | 4 |
| Atoms | 4908 |
| Residues | 634 |
| Bonds (RMSD) | |
| Length (Å) | 0.003 |
| Angles (°) | 0.6 |
| MolProbity score | 2.04 |
| Clash score | 13.98 |
| Ramachandran plot (%) | |
| Outliers | 0 |
| Allowed | 5.75 |
| Favored | 94.25 |
| Rotamer outliers (%) | 0 |
| Cβ outliers (%) | 0 |
| Peptide plane (%) | 0 |
| CaBLAM outliers (%) | 1.46 |
| Box | |
| Lengths (Å) | 72.48, 86.85, 74.44 |
| Angles (°) | 90, 90, 90 |
| Masked dFSC model (0/0.143/0.5) | 3.3/3.5/3.8 |
| Model vs. Data | |
| CC (mask) | 0.80 |
| CC (box) | 0.78 |
| PDB code | 8EY2 |
| EMDB | EMD-28666 |
| EMPIAR | EMPIAR-10810 |

fact that the covalent inhibitor (but not a non-covalent) might induce dimerization in a non-cleaved sample suggests that the trigger of the dimerization is not the N-terminal processing, but the induced fit caused by the covalent linkage itself. This model would also explain why oligomers can be formed by the combination of non-cleaved and cleaved particles rather than only cleaved particles (Supplementary Fig.1).

To confirm this hypothesis, we crystalized M[pro] C145S monomers in the presence of Nirmatrelvir. The crystal structure revealed that despite the C145S mutation, Nirmatrelvir is linked covalently to the protein. Notwithstanding, we also observed displacement of M[pro] helixes αF, αH and αI from adjacent Domain III, which seem to be caused by the steric hindrance from non-cleaved N-terminal amino

acids (Fig. 7). This is like our previous structure of immature M[pro], with the exception that we do not see the displacement of active site residues Phe140, Glu166, Pro168 and Gln189[6]. This suggested that the covalent linkage permit the proper reshaping of the active site, allowing the dimerization event even in the absence of proper N-terminal cleavage. One might infer that in order to prevent dimerization, an inhibitor must be non-covalent and successfully inhibit the monomeric form of M[pro] binding to all the eleven distinct cleavage sites at the viral polyprotein.

If one combines the cryo-EM structure of C145S mutant bound to a uncleaved peptide with our previous X-ray structure of C145S (PDB 7N6N) chains A (cleaved peptide covalently bound to Ser145) and chain B (post-cleaved peptide), we can glimpse the structural modifications of M[pro] during all steps of the enzymatic processing (Fig. 8). These structures reveal that the enzymatic process starts with key amino acids positioned in similar manner with than the apo-structure (Fig. 8a), but loop β166-171 must be pushed towards the enzyme core to accommodate substrate peptides P3-P5 (Fig. 8b). Next, during the covalent enzyme-substrate intermediary, this loop β166-171 must be pushed even further the enzyme core to allow a shorten covalent distance between C/S145 and the scissile-bond carbonyl C and proper anchoring to the active site, while helixes 43-53 and 62-65 show a loosen conformation (Fig. 8c). After cleavage, the product peptide is no more linked to C/S145, allowing the active site domain return to its apo conformation (Fig.8d). During this process, M[pro] active site surface undergoes significant structural and electrostatic potential modifications to accommodate all enzyme-substrate intermediary conformations, now detailed in this series of structures (Fig. 8).

## Discussion

The SARS-CoV-2 M[pro] is a three-domain 34 kDa protein that is essentially active as a dimer[5,6]. Since the first X-ray structure of SARS-CoV-2 M[pro] was deposited at the Protein Data Bank (PDB) under the code 6LU7 in February 2020[23], there are nearly 450 structures up to date from M[pro] available at the PDB up to 1.2 Å resolution (PDB 7K3T), making the M[pro] one of the most well structurally characterized SARS-CoV-2 nsp's[33]. The M[pro] was characterized by X-ray in complex with multiple new drug candidates[5,24–27] and repurposing molecules[24,28,29], fragments from large fragment-screening campaigns[30], as well as endogenous peptides[6,16,31] and peptide-mimetics drug candidates[32]. Moreover, room and cryo temperature X-ray structures and neutron crystallography were successfully used to provide valuable information for detailing the M[pro] mechanism of action[16,33] as well as conformational changes during substrate and ligand recognition[6,17]. Yet, the cryo-EM structure of M[pro] reported here, reveals unique features of the enzyme-substrate recognition and in-solution dynamics.

We also showed that the M[pro] N-terminal cleavage is important for the proper folding of protein, but not critical for the dimerization. Our data suggest that the dimerization event is governed by the induced fit of the protein during the formation of a covalent linkage during cleavage. This likely means that the cleavage of any of the eleven viral M[pro] target sites would serve as a trigger for the dimerization, regardless of the state of the N-terminus. These observations diverge from previous proposed models, in which the cis-cleavage of N and C-terminal between two protomers are necessarily the initial step of the maturation process[13,34]. In fact, previous studies had showed that substrate-induced dimerization can occur in an in-vitro model of the viral polyprotein, where N and C-terminal are bound to other proteins[35]. Our model might also explain why M[pro] studies that used other strategies for protein production that generated authentic N-terminus, such as the cases of MERS-CoV M[pro] where authors opted for using thrombin at the N-termini (therefore, not allowing the protein to perform internal cleavage and complete maturation) had generated protein that behaves mostly as monomers[36]. Curiously, authors have also reported that dimerization of MERS-CoV M[pro] could be triggered

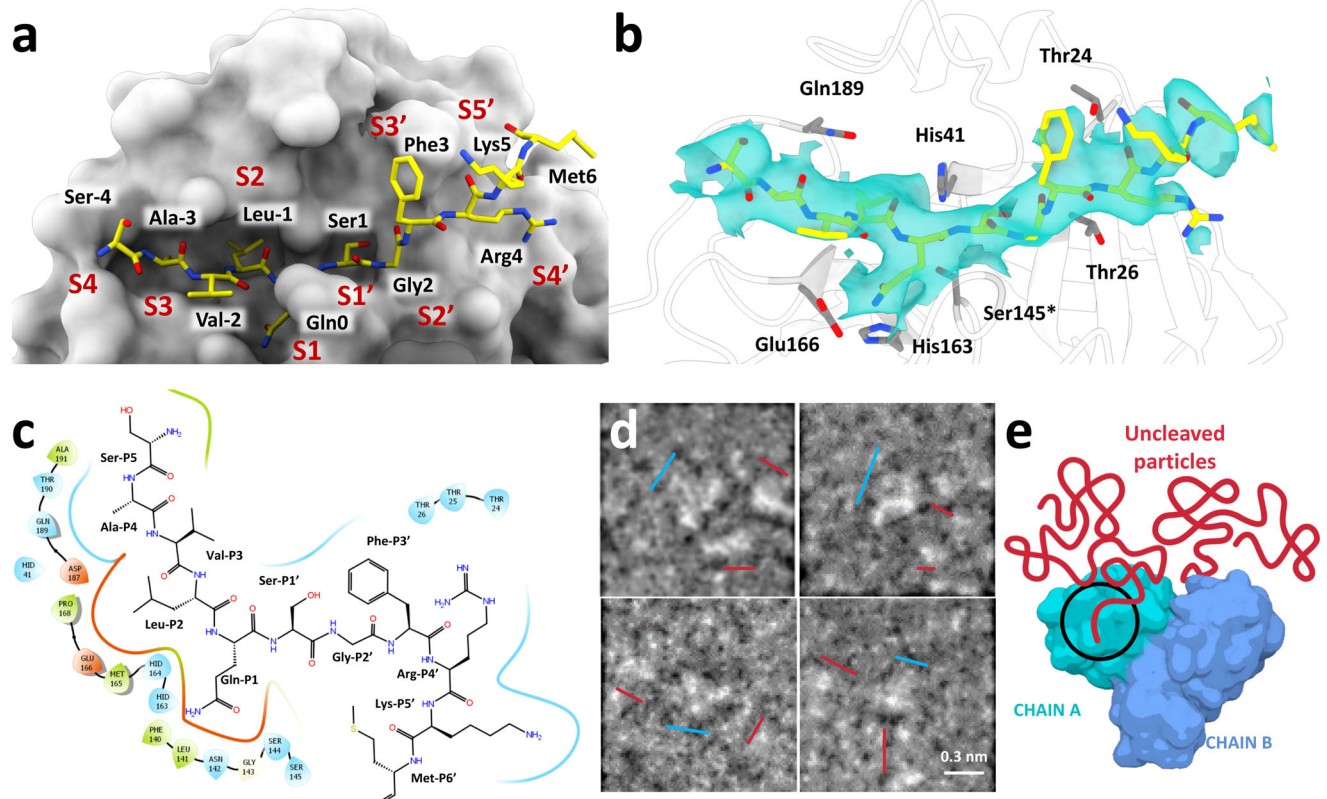

**Fig. 4 | Detailed view on M^pro C145S peptide interaction. a** Active site view of M^pro C145S chain A surface (in grey) bound to nsp4-nsp5 peptide (yellow sticks). Subsites are denotated from S4 to S5′. **b** Detailed view of M^pro C145S chain A active site residues (in grey) bound to nsp4-nsp5 peptide (yellow sticks), with cryo-EM map showed as surface (contour level of 4.55). **c** Interaction scheme between nsp4-nsp5 peptide and M^pro C145S chain A. **d** Selected low-pass filtered particles, highlighting dimer particles (marked with a blue line) bound to monomeric uncleaved particles (marked with a red line). Scale bar is show at the bottom left. **e** Schematic representation of M^pro C145S dimer (blue) bound to uncleaved particles (red).

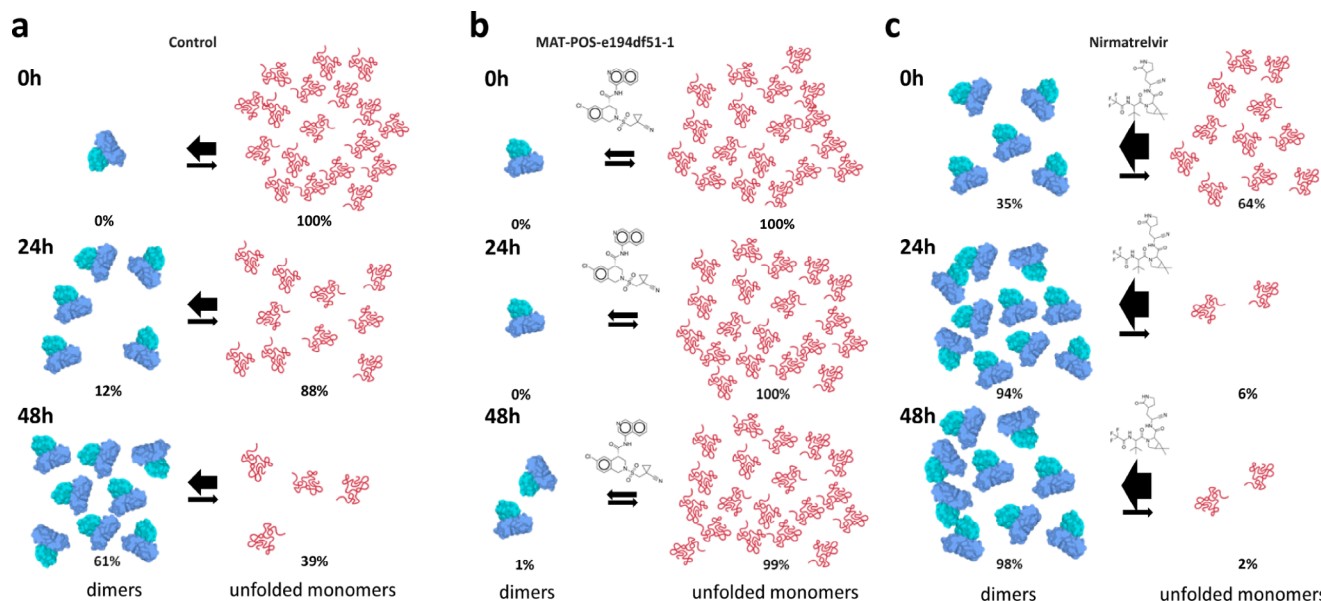

**Fig. 5 | Schematic representation of in solution dynamics of SARS-CoV-2 C145S M^pro monomeric form (sample 1) analyzed with SEC-MALS. a** Control reaction containing monomers at 0 h (top), after 24 h incubation (middle) and after 48 h (bottom). **b** Monomers conversion reaction in presence of non-covalent inhibitor MAT-POS-e194df51-1 at 0 h (top), after 24 h incubation (middle) and after 48 h (bottom). **c** Monomers conversion reaction in presence of covalent inhibitor Nirmatrelvir at 0 h (top), after 24 h incubation (middle) and after 48 h (bottom).

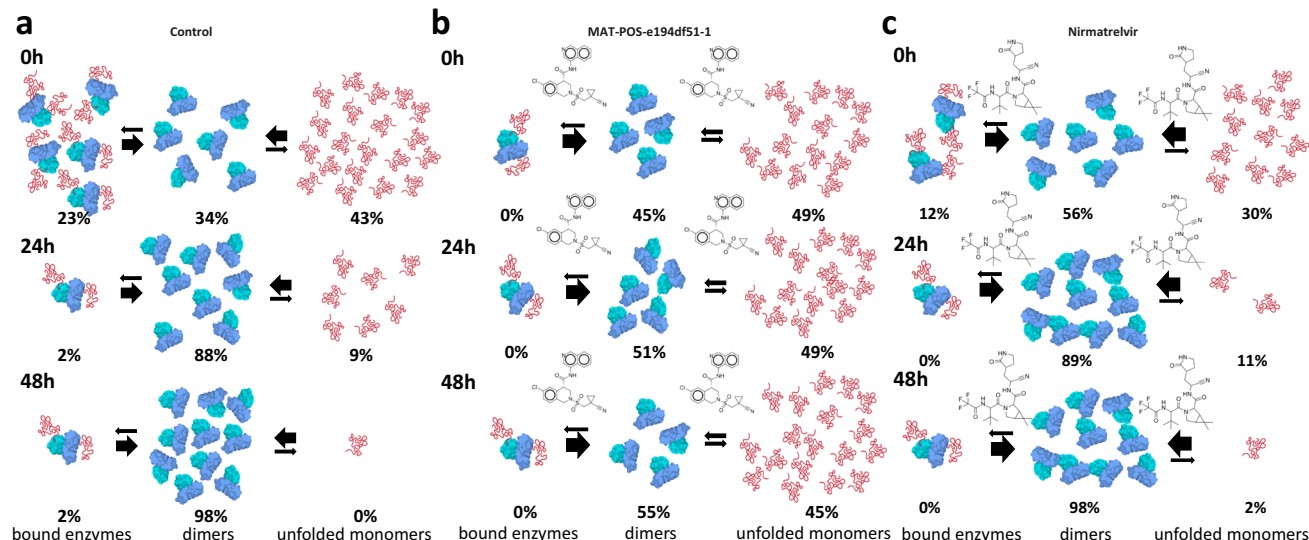

**Fig. 6 | Schematic representation of in solution dynamics of SARS-CoV-2 C145S M<sup>pro</sup> tetrameric form (sample 2) analyzed with SEC-MALS. a** Control reaction containing tetramers at 0 h (top), after 24 h incubation (middle) and after 48 h (bottom). **b** Tetramers conversion reaction in presence of non-covalent inhibitor MAT-POS-e194df51-1 at 0 h (top), after 24 h incubation (middle) and after 48 h (bottom). **c** Tetramers conversion reaction in presence of covalent inhibitor Nirmatrelvir at 0 h (top), after 24 h incubation (middle) and after 48 h (bottom).

by substrate, which also agrees with our proposed mode[36]. Still, more data is required to conclude nascent M<sup>pro</sup> are processed and converted into mature, and if this process is dependent of processing event, the processed particle or a combination of both. These observations impact our understanding of the maturation process of coronaviruses M<sup>pro</sup>, as well as the pharmacodynamics differences between covalent and noncovalent inhibitors.

## Methods
### Cloning and protein production
Details of construct planning, cloning and protein expression are given elsewhere[6]. Briefly, the viral cDNA template (GenBank MT126808.1), kindly provided by Dr. Edison Durigon (University of São Paulo, São Paulo, Brazil), was used for cloning the coding region of coding region of M<sup>pro</sup> (residues 3264-3569) using primers: Fw 5' CAGGGCGCCCAT-GAGTGGTTTTAGAAAAATGGCATTC 3' and Rv 5' GACCCGACGCGGT-TATTGGAAAGTAACACCTGAGAC 3'. This sequence was inserted in pET_M11 vector using the LIC method[37]. This plasmid was then used as the template for the insertion of M<sup>pro</sup> native N-terminal residues (GAMSAVLQ ↓ SGFRK) by inverse PCR using primers: Fw: 5' GCTGCA-GAGTGGTTTTAGAAAAATGGCATTC 3' and Rv: 5' ACGGCTGA-CATGGCGCCCTGAAAATA 3'. Then, C145S mutation was inserted in this plasmid by inverse PCR using primers Fw 5' CCTTAATGGTT-CATCTGGTAGTG 3' and Rv 5' AATGAACCCTTAATAGTGAAATTGG 3', resulting in the pET_M11-C145S-M<sup>pro</sup>. This construct contains a N-termini 6x histidine tag followed by a TEV cleavage site, and the nsp4 C-termini recognition sequence amino acids (Ser-4, Ala-3, Val-2, Leu-1, Gln-0 ↓) followed by the M<sup>pro</sup> sequence containing C145S substitution[6].

For protein production, pET_M11-C145S-M<sup>pro</sup> was used to transform *E. coli* BL21 cells and cultured in ZYM-5052 media[38] at 37 °C and 200 RPM to an $OD_{600}$ of 0.8, followed by expression at 18 °C, 200 RPM for 16 h. Cells were harvested by centrifugation at 5000 *g* for 40 min at 4 °C, resuspended in lysis buffer (20 mM Tris pH 7.8, 150 mM NaCl, 1 mM DTT) and disrupted by sonication. Lysate containing soluble protein was clarified by centrifugation at 12,00x *g* for 30 min at 4 °C.

C145S M<sup>pro</sup> was purified by immobilized metal chromatography (IMAC) using a 5 mL HisTrap FF column (GE Healthcare). After column washing with buffer A (20 mM Tris pH 7.8, 150 mM NaCl, 25 mM Imidazole), the protein was eluted with buffer A supplemented with

250 mM imidazole. Sample was buffer exchanged using a 5 mL HiTrap desalting column (GE Healthcare) equilibrated with buffer A. To remove the 6xHis-tag, 2 mg of TEV protease and 4 mM DTT were added to the sample and incubated overnight at 4 °C. Next day, non-cleaved protein and TEV were removed by a second step of IMAC in buffer A. The protein was then purified by size-exclusion chromatography (SEC) using a HiLoad 16/60 Superdex 75 column (GE Healthcare) equilibrated with gel filtration buffer (20 mM Tris pH 7.8, 150 mM NaCl, 1 mM EDTA, 1 mM DTT). The SEC profile of this sample showed multiple overlapping peaks of mixed oligomeric states of M<sup>pro</sup>, like previously described[6]. Peaks with higher retention time are mainly composef (sample 1), while peaks with lower retention time are mainly composed by tetramers (sample 2). Protein purity was analyzed by SDS-PAGE and quantified using the theorical extinction coefficient of 32,890 M<sup>-1</sup>.cm<sup>-1 39</sup>. Fresh purified fractions of both peaks were aliquoted at 10 mg.mL<sup>-1</sup> and immediately flash-frozen in liquid nitrogen.

### Grid preparation
For grid preparation, freshly purified C145S M<sup>pro</sup> samples from the tetrameric of SEC peak (sample 2) were thawed in ice, spined and diluted in gel filtration buffer. Then, 3 μL of 0.25 mg.ml<sup>-1</sup> of samples was applied on glow discharged Quantifoil 300 mesh Cu R2/2 grid. Grid was blotted using Vitrobot (FEI) for 2.5 s with blot force 1 at humidity 100% and 4 °C before plunge freezing into liquid ethane.

### Cryo-EM data acquisition
Data were collected on Titan Krios operated at 300 kV using a K3 detector in counted super-resolution mode with slit width of 20 eV at a nominal magnification of 130k x corresponding to a calibrated pixel size of 0.653 Å at the specimen level. Movies were recorded over 1 sec at dose rate of 18.5 e-/px/s and fractionated into 43 frames. Data acquisition was done using ThermoFisher Scientific EPU 2.12 with a defocus range of −0.5 to −2.5 μm.

### Cryo-EM data processing
For high-resolution structure determination, the 13,770 super-resolution movies collected at Titan were dose-weighted, aligned and two times binned using MotionCor2[40,41] implemented in RELION v3.1[41], resulting in a physical pixel of 0.653 Å. The contrast-transfer-

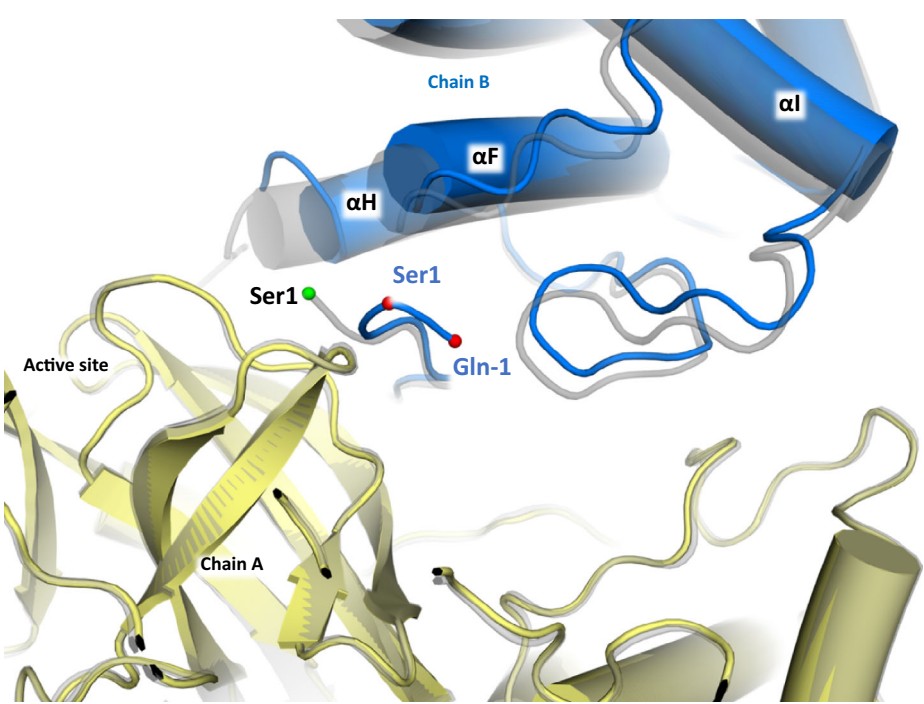

**Fig. 7 | Cartoon model of X-ray structure of M$^{pro}$ C145S bound to Nirmatrelvir, with chain A showed in yellow, and chain B showed in blue.** Ser1 and Gln-1 alpha carbons are highlighted as red spheres. Native M$^{pro}$ is shown as grey transparent cartoon, with Ser1 alpha-carbon highlighted as a green sphere.

function (CTF) of resulting micrographs was estimated using CTFFIND-v4.1[42].

1,956,496 particles were picked using SPHIRE-crYOLO[43] and down sampled by a factor of 3, extracted with a box size of 72 px. Particles were filtered and sorted with 3 rounds of 2D classification. 569,725 good particles were selected for De novo 3D model. Particles were re-extracted with full resolution. Particles were further filtered and sorted with 3D classification. The particles from the best 3D class were selected for 3D refinement and Bayesian polishing in Relion v3.1[44].

Refine3D volume and the 227,534 shiny particles were then imported to cryoSPARC v3.2.0 for refinement[45]. For the deposited structure, map was initially refined using Homogeneous Refinement algorithm with 1 extra final pass, 7 Å initial lowpass resolution and C2 applied symmetry, resulting in a refined map with gold-standard Fourier shell correlation (gsFSC) of 3.75 Å[46]. The resulting map was used for the Non-uniform Refinement[47] with 1 extra final pass, 8 Å initial lowpass resolution, C2 applied symmetry, per-particle defocus optimization and per-group CTF optimization, resulting in a refined map with gsFSC of 3.6 Å. The resulting map was than refined using Local Refinement algorithm with C2 imposed symmetry, resulting in a map with gsFSC of 3.5 Å. The same protocol was also tested with C1 symmetry, resulting in a nearly identical map with gsFSC of 3.76 Å. Therefore, the C2 map was used for further modelling and refinement. Map resolution was calculated using cryoSPARC Local Resolution and 3DFSC Processing Server[48]. Statistics of data collection, processing and model refinement for the C2 are available in Table 1. Schematic of the data processing steps is available in Fig. 2.

## Modeling, refinement and analysis
Initial docking of model into map was performed using PDB 7N6N and sharpened map from cryoSPARC using UCSF ChimeraX v1.2.5[49]. For sharpening and enhance visualization of the high-resolution map, we used phenix auto-sharpen[50]. Model building and refinement was conducted with Phenix Real Space Refinement[51] and Coot[52], and validation was conducted with MolProbity[53]. Figures were prepared using PyMOL and ChimeraX v1.2.5. Statistics of model refinement are available in Table 1.

## 3D variability analysis
To analyze variability of SARS-CoV-2 M$^{pro}$ C145S particles, we used the 3D Variability Analysis (3DVA) tool available in cryoSPARC v3.2.0[54]. For that, the 569,725 full size particles used during cryo-EM data processing 3D classification (in Relion) were imported in cryoSPARC v3.2.0. These particles and Refine3D volume were incorporated in a symmetry-free homogeneous refinement cycle. Then, particles and mask were used to compute 4 eigenvectors of the covariance matrix of the data distribution using 3DVA[54], with resolution filtered at 5 Å. Model series were generated using 3DVA display tool. Images were generated with ChimeraX v1.2.5.

## Denaturing electrospray mass spectrometry (ESI-TOF) intact mass analysis
Reversed-phase chromatography was performed in-line prior to mass spectrometry using an Agilent 1100 HPLC system (Agilent Technologies inc. – Palo Alto, CA, USA). One sample of freshly prepared C145S containing tetrameric peaks (sample 2) was diluted to 20 μg/ml in 0.1% formic acid, and fifty μl was injected on to a 2.1 mm × 12.5 mm Zorbax 5 μm 300SB-C3 guard column housed in a column oven set at 40 oC. The solvent system used consisted of 0.1% formic acid in LC-MS grade water (Millipore, solvent A) and 0.1 % formic acid in methanol (LC-MS grade, Chromasolve, solvent B). Chromatography was performed as follows: Initial conditions were 90 % A and 10 % B and a flow rate of 1.0 ml/min. After 15 s at 10 % B, a two-stage linear gradient from 10 % B to 80 % B was applied, over 45 s and then from 80% B to 95% B over 3 s. Elution then proceeded isocratically at 95 % B for 1 min 12 s followed by equilibration at initial conditions for a further 45 s. Protein intact mass was determined using an MSD-ToF electrospray ionisation orthogonal time-of-flight mass spectrometer (Agilent Technologies Inc. – Palo Alto, CA, USA). The instrument was configured with the standard ESI source and operated in positive ion mode. The ion source was operated with the capillary voltage at 4000 V, nebulizer pressure at 60 psig, drying gas at 350$^{o}$C and drying gas flow rate at 12 L/min. The instrument ion optic voltages were as follows: fragmentor 250 V, skimmer 60 V and octopole RF 250 V.

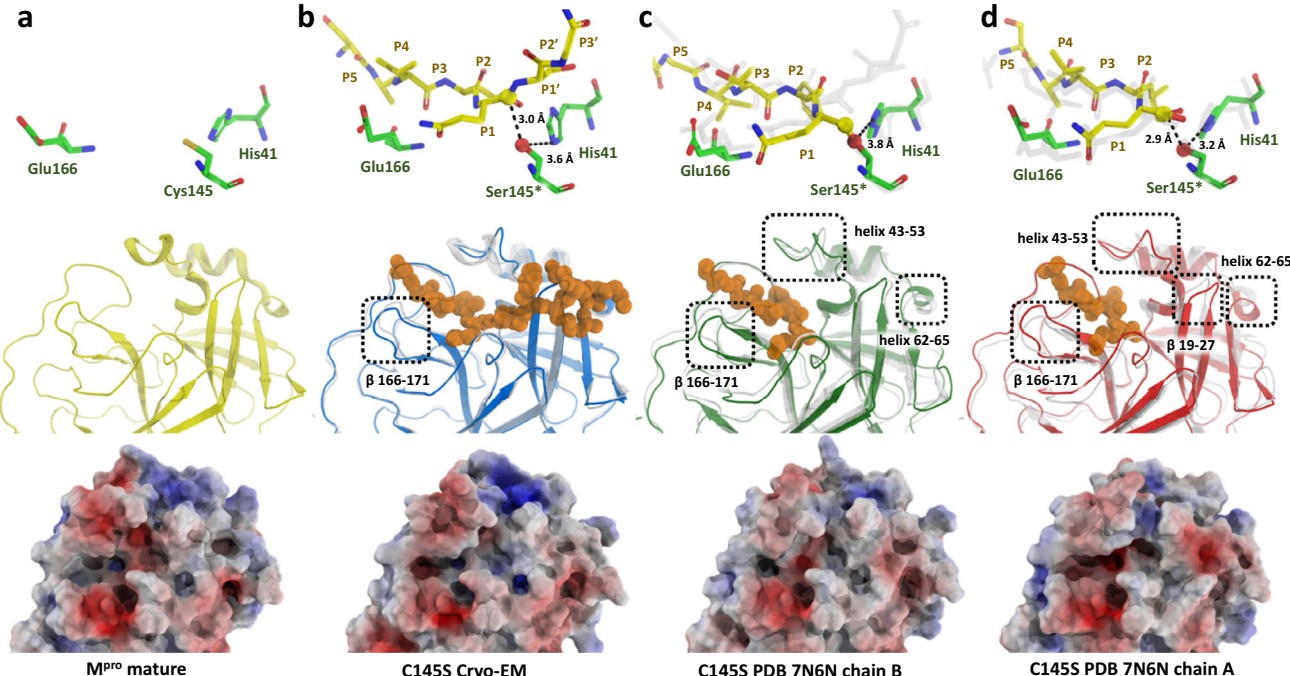

**Fig. 8 | Active site comparison between apo and intermediary states of M^pro.**
**a** Key active site residues (green sticks) of M^pro in apo form (top), cartoon view of active site in apo state in yellow (middle) and calculated electrostatic potential projected into surface of M^pro active site (bottom). **b** Key active site residues (green sticks) of M^pro C145S bound to intact peptide (top), cartoon view of active site from the respective form (middle) and calculated electrostatic potential projected of respective form (bottom). **c** Key active site residues (green sticks) of M^pro C145S covalently bound to cleaved peptide forming the enzyme-substrate intermediary complex (top), cartoon view of active site from the respective form (middle) and calculated electrostatic potential projected of respective form (bottom). **d** Key active site residues (green sticks) of M^pro C145S in complex with post-cleaved peptide (top), cartoon view of active site from the respective form (middle) and calculated electrostatic potential projected of respective form (bottom). The transparent sticks and cartoons (grey) in the top and middle figures represent the structural position from the relative elements of the previous step.

## Native electrospray mass spectrometry (ESI-TOF) intact mass analysis

For native mass spectrometry, one sample of freshly prepared sample 2 were held on ice and buffer exchanged into 75 µl of 50 mM ammonium acetate pH 7.5 by 3 rounds of gel filtration using BioGel P6 (Biorad) spin columns according to the manufacturer's instructions. Mass spectra were acquired using an Agilent 6530 QTOF operating in positive ion 1 GHz mode using a standard ESI source. Samples were introduced via a syringe pump at a flow rate of 6 µl/min. The ion source was operated with the capillary voltage at 3500 V, nebulizer pressure at 17 psig, drying gas at 325°C and drying gas flow rate at 5 L/min. The instrument ion optic voltages were as follows: fragmentor 430 V, skimmer 65 V and octopole RF 750 V. This same setup was used to describe the tetrameric native form of *E. coli* LacZ (theorical mass of 465,932 Da), with observed folded mass of 466,092 Da for tetramer[55].

## In solution oligomeric state analysis

In solution oligomeric states of the purified samples were evaluated by size exclusion chromatography coupled with multi-angle light scattering (SEC-MALS). All assays were performed in running buffer composed by 20 mM Tris-HCl pH 7.8, 100 mM NaCl and 1 mM DTT, as described previously[6]. For that, one sample of 50 µL of each M^pro C145S oligomers at concentration of 50 µM were injected in a Waters 600 HPLC system (Waters) coupled in-line with an UV detector, a miniDAWN TREOS multi-angle light scattering apparatus (Wyatt Technology), a column Superdex 200 Increase 10/300 GL (GE Healthcare) with a flow rate of 0.5 mL/min, and a refractive index detector Optilab T-rEX (Wyatt Technology). The light-scattering detectors were normalized with bovine serum albumin (Sigma-Aldrich). Data were collected and analyzed with the ASTRA 7 integrated software, provided by Wyatt.

Samples analyzed contained 100 µg of C145S M^pro monomers or tetramers that were incubated with 1% DMSO (control) or ligand for 0 h, 24 h, or 48 h. Non-covalent inhibitor MAT-POS-e194df51-1, obtained from Moonshot COVID Consortium was tested in the inhibitor:protein ratio concentration of 4:1 and 0.4:1, and incubated for 0 h, 24 h, or 48 h incubation with each sample. Covalent inhibitor Nirmatrelvir was tested in the inhibitor:protein ratio concentration of 4:1, and incubated for 0 h, 24 h, or 48 h. Given the lack of sufficient resolution of the methods, trimers and tetramers peaks were summed and treated as tetramers. All generated data is available at Supplementary Fig. 3.

## Crystallization of and data analysis C145S monomers with Nirmatrelvir

Monomers samples of C145S M^pro (sample 1) at 5 mg/mL were incubated with Nirmatrelvir in a 1:5 protein:compound ratio for 1 h at room temperature. Then, sitting drop crystallization plates were set using in condition 0.1 M MES pH 6.7, 5% DMSO, 8% PEG 4000 at 20 °C. Crystals were cryo-protected with 30% PEG 400, and data was collected at MANACA beamline (SIRIUS). Data was processed with XDS via Autoproc[56,57]. Data was scaled using Aimless via CCP4[58], and molecular replacement was performed with Phaser[59], using PDB 7BMG as template. Data collection statistics are available in Table 2. Data was modelled with Coot and refined with Acedrg and Refmac[52,60–63]. Figures were made with ChimeraX, Pymol and BioRender.com. Electron density of this structure is visible in Supplementary Fig. 7.

## Native electrospray mass spectrometry using Orbitrap Q-Exactive UHMR

For a confirmatory experiment using a different setup, native mass spectrometry was carried out using an Orbitrap Q-Exactive UHMR as

## Table 2 | X-ray crystallography data collection, refinement and validation parameters and statistics

| | M$^{pro}$ C145S with Nirmatrelvir |
|---|---|
| *Data collection* | MANACA beamline |
| Space group | $P2_12_12_1$ |
| Cell dimensions | |
| *a,b, c* (Å) | 67.64, 101.96, 103.62 |
| *α, β, γ* (°) | 90.0, 90.0, 90.0 |
| Resolution range (Å) | 72.75 - 1.74 |
| Unique Reflections | 53189 (267) |
| Multiplicity | |
| Completeness (%) | 71.5 (3.6) |
| *I/σI* | 8.9 (1.4) |
| $R_{p.i.m.}$ (%)[a] | 0.07 (0.476) |
| $CC_{1/2}$[b] | 0.995 (0.556) |
| *Refinement* | |
| $R_{work}$/$R_{free}$[c] | 0.193/0.234 |
| Number of atoms | 5489 |
| Waters | 713 |
| Ligands | 70 |
| Protein residues | 4706 |
| RMS(bonds) (Å) | 0.0138 |
| RMS(angles) (°) | 1.80 |
| Ramachandran favored (%) | 96.35 |
| Ramachandran outliers (%) | 0 |
| Clashscore[d] | 5.97 |
| Average *B*-factors (Å²) | 26.31 |
| Macromolecules | 24.92 |
| Ligands | 22.62 |
| Solvent | 37.23 |
| PDB code | 8EYJ |

[a]$R_{p.i.m.} = \sum_{hkl} \{1/[N(hkl) - 1]\}^{1/2} \times \sum_j | I_j(hkl) - \langle hkl \rangle | / \sum_{hkl} \sum_j I_j(hkl)$[61].
[b]$CC_{1/2}$ is the correlation coefficient determined by two random half data sets[62].
[c]$R_{work} = \sum_{hkl} | F_o(hkl) - F_c(hkl) | / \sum_{hkl} F_o(hkl)$. $R_{free}$ was calculated for a test set of reflections (5%) omitted from the refinement.
[d]Clashscore is the number of clashes calculated for the model per 1000 atoms[53].

previously described[64]. Briefly, analyte solutions were buffer exchanged into 200 mM ammonium acetate pH 7.4 (Sigma Aldrich) using Zeba micro spin desalting columns (7 kDa MWCO, Pierce). One sample containing 1-3 µL of analyte (in 200 mM ammonium acetate) was loaded into in-house prepared gold-coated glass capillaries pulled to ~1 µm point. The capillary was biased 1.0 – 1.2 kV relative to the heated metal capillary (100 °C) to generate positive ions by nanoelectrospray ionization. The mass spectrometer was operated in the positive ion mode using the manufacturers' recommended settings for "low m/z" detection. The following parameters were manually adjusted: in-source trapping potential: 20 – 50 V, 4 ms; HCD potential: 1 – 10 V; pressure setting: 8.5. Native mass spectra were visualized in QualBrowser (Thermo Fisher Scientific) and OriginPro 2021 (OriginLab Corporation, Northampton, MA). M/z spectra here and from native ESI-TOF were analyzed using Masshunter B.07.00 (Agilent); ESIprot[65] and using an ion table. Charge states were assigned manually. Results of this run are available in Supplementary Fig. 6.

### Reporting summary
Further information on research design is available in the Nature Portfolio Reporting Summary linked to this article.

## Data availability
All data associated with this study are publicly available. All collected movies and polished particles used for final structure determination are available at Electron Microscopy Public Image Archive (EMPIAR) under the code EMPIAR-10810. Cryo-EM maps and structural models are available at Protein Data Bank (PDB) under code 8EY2 and Electron Microscopy Data Bank (EMDB) under the code EMD-28666, and the X-ray model is available under PDB code 8EYJ. The structures used in this study are available in PDB database under the codes 7N5Z, 7KPH, 7N6N, 7K3T and 7DVP. A source data file is provided with this paper for all SEC-MALS and native mass spectroscopy. Source data are provided with this paper.

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

## Acknowledgements

Authors acknowledge Diamond Light Source for access and support of the cryo-EM facilities at the UK national electron Bio-Imaging Centre (eBIC) through proposal BI27083 and NT29349, funded by the Wellcome Trust, MRC and BBSRC. Authors acknowledge COVID Moonshot Consortium for supplying compound and Prof. Carlos Alberto Montanari

for kindly supplying Nirmatrelvir. ASG acknowledge Andressa Patricia Alves Pinto for support with SEC-MALS. This project was funded by Coordenação de Aperfeiçoamento de Pessoal de Nível Superior (CAPES – Project 88887.516153/2020-00, ASG) and Fundação de Amparo à Pesquisa do Estado de São Paulo (FAPESP projects 2013/07600-3, 2015/16811-3 and 2016/19712-9, GO). This work has received support from the EU/EFPIA/OICR/McGill/KTH/Diamond Innovative Medicines Initiative 2 Joint Undertaking (EUbOPEN grant n° 875510, RC). This research used facilities of the SIRIUS, part of the Brazilian Center for Research in Energy and Materials (CNPEM), a private non-profit organization under the supervision of the Brazilian Ministry for Science, Technology, and Innovations (MCTI). The MANACA beamline staff is acknowledged for the assistance during the experiments of proposal 20220605. Research reported in this publication was supported in part by NIAID of the National Institute of Health under award number U19AI171399. The content is solely the responsibility of the authors and does not necessarily represent the official views of the National Institute of Health. Research in the Robinson laboratory is supported by a Medical Research Council (MRC) program grant (MR/V028839/1). This research was also funded in part, by the Wellcome Trust, Grant No. 221795/Z/20/Z. TJE is a Junior Research Fellow at Linacre College and was supported by a Royal Society Newton International Fellowship. TJE and CVR are grateful for generous support provided by the University of Oxford COVID-19 Research Response fund and its donors (BRD00230).

## Author contributions

A.S.G. conceived this project. G.D.N. performed biochemical and biophysical experiments. Y.S. prepared grids and data collection. A.S.G. and Y.S. processed the cryo-EM data. A.S.G. and G.D.N. performed modelling and analysis. L.K., H.E., R.C., T.H.E. and C.V.R. performed MS experiments. A.S.G. wrote and conceived this manuscript. A.S.G., R.S.F., Y.S., C.D.O. and G.O. revised this manuscript.

## Competing interests

The authors declare no competing interests.

## Additional information

## The COVID Moonshot Consortium

Gabriela Dias Noske[1,6], Rafaela Sachetto Fernandes[1], Glaucius Oliva [1], Andre Schutzer Godoy[1] ✉, C. David Owen[2] & Lizbé Koekemoer[3]

A full list of members and their affiliations appears in the Supplementary Information.

