## [Peer review file · Nature Communications]

REVIEWER COMMENTS

Reviewer #1 (Remarks to the Author):

The main innovation of this manuscript is that that oligomeric states of Mpro are not dependent of N-terminal processing, which the authors claim is in opposition to prevailing expert opinion. They work with an inactive (or slow) active site variant, Cys145S, drawing conclusions from this. This manuscript was not written clearly from this reviewer's point of view, making it very difficult to review. It is possible that some of the concerns raised below could be addressed in a work that is rewritten to enumerate the rationale for the study design together with a diagram illustrating the key steps in maturation. I do not recommend publication of this manuscript in its current form.

Major concerns:

- 1) The literature review appears to be incomplete and provides few references, other than those of the authors, to studies and structures that support or refute the claim that the oligomeric states of Mpro are not dependent upon N-terminal processing.
- 2) The authors published (2021) an X-ray study featuring the same CysSer mutant studied in this manuscript. They do not clearly enumerate the major differences and similarities between their new EM structure and old X-ray structure. They describe the EM structure in great detail, but the analysis of these details into a coherent structural framework is lacking.
- 3) The terminology should be more precise. "Oligomeric forms" would apply to the dimer, trimer, and tetramer. The authors use "oligomeric" when only "dimeric" would justify the conclusions they draw.
- 4) The native MS analysis has serious flaws, to the extent that the conditions used were probably not native. Most native MS studies (of oligomers) are performed using nanoflow ionization (at very low flow rates and very small internal diameter tips) and higher concentration buffers (100-200 uM). This enables the use of gentle desolvation, e.g. lower capillary temperatures and lower voltages, i.e., in regions of hypersonic expansion that can dissociate oligomers and unfold proteins. The high flow, high temperature, high voltage conditions used by the authors likely result in dissociation of the tetrameric forms of Mpro to smaller trimers, dimers, and monomers (and likewise the dissociation of the trimeric form). This could lead to the authors misinterpreting a central finding of this manuscript- that mixed dimers of unprocessed and processed (cleaved) Mpro exist in solution. In fact, these mixed dimers could be artifacts of the dissociation/reassociation in the gas phase. Additionally, the unfolded monomer observed in the native MS are consistent with unfolding within the mass spectrometer. It should be noted that mixed tetramers were observed as the authors claim, and if such a mixed tetramer can only exist in the new model proposed here, that is evidence to support their claims. In summary the native MS is more consistent with the prevailing model than the new model proposed here. The authors note this as well "It is also clear by the mass relative quantities that the equilibria 96 of oligomeric states favor the states where more cleaved elements are present (Fig. 1 97 and S1), still befitting with that model were N-terminal cleavage is directly involved in the 98 oligomerization." To prove native

conditions under their unusual experimental conditions an oligomeric control such as aldehyde dehydrogenase or GROEL would be necessary.

5) The authors do not give sufficient rationale for their experimental design. Why is it important that a covalent ligand (which we must assume was intended to bind Cys145) binds C145S Mpro? Couldn't such a ligand be binding off target, and if so, where is it binding? It is unclear why this system (pertaining to the last third of the manuscript) should be taken as representative of wild-type Mpro.

6) Confirmation of some results with wild-type Mpro would be welcome.

7) Both the native MS and cryoEM results seem to favor the prevailing model, rather than the novel model proposed here.

Minor comments:

"To obtain heterologous mature Mpro" should be reworded. Do the authors mean "heterologously expressed, mature Mpro?"

"The final model showed none rotamers" Do the authors mean "nine rotomers"?

Reviewer #2 (Remarks to the Author):

This is a potentially significant study on the oligomeric state of the SARS-COV-2 main protease Mpro in solution, and the effect of inhibitors in this process, and the possible involvement of N-terminal processing. The main finding of their study is that the key dimerization process is not triggered by N-terminal processing, but by substrate processing.

The paper is well researched and clearly described. My only real concern is the choice of techniques that were used by the Researchers, particularly as important potentially concentration dependent associative processes were being studied and conclusions being drawn thereof. Mass spectroscopy is not a solution method and whereas SEC-MALS is a very powerful solution method for the analysis of polydisperse and heterogeneous systems that are not concentration dependent, the necessity of a separation matrix and issues of concentration control creates difficulties for concentration dependent reversible processes. How do dilutions on the columns affect the oligomerization behaviour? Only one injection concentration seems to have been attempted. I would have greater confidence in their conclusions, particularly based on Fig 5 if they had been reinforced by different injection concentrations, and more importantly also if the results had been supported by measurements from the analytical ultracentrifuge.

A revised paper with these critical additional measurements would be more useful.

Reviewer #3 (Remarks to the Author):

SARS-CoV2 main protease is a promising drug target proven by the matured drug trials targeting this protein. Understanding the maturation process and elucidating the conformational changes associated with every steps will help to develop more powerful therapeutics.

With this goal in mind, Noske et al have presented this manuscript providing a characterisation of the protein by an ensemble of techniques including Native mass spectrometry, cryo-EM and crystallography. The main points are

- i) capturing the structure of the dimeric protein bound to its uncleaved substrate. This structure, although stands against the existing model, wherein maturation of the protease by its self cleavage is crucial for its dimerisation to form active protease, it complements the missing information that helps to understand the maturation process better. Along with publications from other groups, the current author's earlier paper <https://doi.org/10.1016/j.jmb.2021.167118>, shows that any presence of additional residues in its N-terminus leads to monomeric and less active enzyme called immature MPro. In this line, the finding here sheds new insights into the maturation process of the main protease.
- ii) Experimental proof of the equilibria shifting to dimeric form while bound to a covalently linked inhibitor and to monomeric while bound to a non-covalently bound inhibitor will be exploited for designing better therapeutics.

The study is well planned and sufficient experimental proofs have been given. I recommend the acceptance of the manuscript. Some important points to be looked into before publication are:

1. The cryo-EM model has some clashes, especially, between the peptide and the main protease. This needs to be addressed and corrected and accordingly the protein-peptide interaction details need to be amended in the manuscript. Also, clashes at the interface, if any, needs to be addressed.
2. The 8dg3 model has some ramachandran outliers, which needs to be corrected as well.
3. It is mentioned beyond Met6 the density is not visible in the cryo-EM structure. However, it is not clear how many residues are invisible? This detail should also be included in the cloning part.
4. The proof shown from raw low-pass filtered particles for Mpro monomeric unfolded particles (Fig 4D) seem very weak. There is a possibility that these particles could be formed during vitrification due to air-water interface. As the authors could not produce a 2D averages of these due to the mobility, statistics in the form of percentage of these particles in this data set could help. Also, there is no mentioning of how long after purification the cryo-EM was performed, for ex, right after purification indicating the sample at 0h, as to judge whether lots of these unfolded particles are anticipated.

Minor remarks:

1. It should be mentioned that the surface shown in Fig3A is from the cryo-EM reconstruction or generated from the 3D model fitted in the cryo-EM map.
2. The manuscript should be thoroughly read for consistency in NSP or nsp notations, repetition of words, spelling errors, etc. I found many but could list only some below
3. Line 126 RMSD to 3 decimals not needed.
4. Line: 170 Leucine and valine: "e" is missing.
5. Fig 5D should be Fig 6C
6. Line 273: should be "with the exception"
7. Line 274: Gln189 "6" ?

REVIEWER COMMENTS

Reviewer #1 (Remarks to the Author):

The main innovation of this manuscript is that that oligomeric states of Mpro are not dependent of N-terminal processing, which the authors claim is in opposition to prevailing expert opinion. They work with an inactive (or slow) active site variant, Cys145S, drawing conclusions from this. This manuscript was not written clearly from this reviewer's point of view, making it very difficult to review.

It is possible that some of the concerns raised below could be addressed in a work that is rewritten to enumerate the rationale for the study design together with a diagram illustrating the key steps in maturation.

I do not recommend publication of this manuscript in its current form.

Major concerns:

1) The literature review appears to be incomplete and provides few references, other than those of the authors, to studies and structures that support or refute the claim that the oligomeric states of Mpro are not dependent upon N-terminal processing.

A: Thank you for the comments. We included more references in the second paragraph of discussion section. We presented similar studies that have been done using a mutated version for SARS1 Mpro and found similar results, as well as studies showing that a in-vitro model of the polyprotein still dimerizes in presence of substrate. These modifications are depicted across lines 333-345

2) The authors published (2021) an X-ray study featuring the same CysSer mutant studied in this manuscript. They do not clearly enumerate the major differences and similarities between their new EM structure and old X-ray structure. They describe the EM structure in great detail, but the analysis of these details into a coherent structural framework is lacking.

A: Thank you for the comments. In the `old` xray structure, we saw the active site of chain A bound to N-terminal peptide post cleavage. In that structure, nsp4 aa Ser-4, Ala-3, Val-2, Leu-1, Gln-0 were found bound to positive sites of Mpro, probably due to the slow cleavage process during the crystallization setup. In here, cryo-EM structure of freshly prepared C145S bound to N-terminal reveals details of the pre-cleavage process, which serves as a model for the initial step of the maturation process. Our model was better explained in this new version and validate with extra native ms data.

3) The terminology should be more precise. "Oligomeric forms" would apply to the dimer, trimer, and tetramer. The authors use "oligomeric" when only "dimeric" would justify the conclusions they draw.

A: Thank you for your suggestions. We followed the terminology proposed by the reviewer to avoid redundancy. These modifications are depicted in the abstract, and lines 295, 330-345,

4) The native MS analysis has serious flaws, to the extent that the conditions used were probably not native. Most native MS studies (of oligomers) are performed using nanoflow ionization (at very low flow rates and very small internal diameter tips) and higher concentration buffers (100-200 uM). This enables the use of gentle desolvation, e.g. lower capillary temperatures and lower voltages, i.e., in regions of hypersonic expansion that can dissociate oligomers and unfold proteins. The high flow, high temperature, high voltage conditions used by the authors likely result in dissociation of the tetrameric forms of Mpro to smaller trimers, dimers, and monomers (and likewise the dissociation of the trimeric form).

A: Thank you for the comments and suggestions. Using the method we describe, we observed an ion distribution for MPro monomer in its folded form which is clearly distinct from the smaller distribution of the unfolded (denatured monomer). The charge separation and intensity distribution of the larger monomer is characteristic of protein in the native condition, whereas the smaller monomer is characteristic for the denatured condition. These clear differences form the basis for interpretation of all native MS results however they are produced. The protein complexes observed with stoichiometry consistent with our other observations and

predicted by the MPro activation mechanism is difficult to explain as anything other than native structures. We believe our native MS conclusions are unequivocal and self-evident.

Moreover, the conditions we use maintain the native state. Reviewer number 1 is correct that most laboratories use nanoflow and the static infusion method for acquisition of native spectra. However their understanding of the ESI process is incomplete. Complete desolvation can be achieved using counter-flow drying gas at high temperature. During this desolvation, latent heat of evaporation is drawn from the electrospray droplet itself causing cooling and maintaining the temperature within droplet close to ambient, thus maintaining native conditions. The unequivocal evidence for this is that functionally and structurally correct proteins and complexes are routinely observed this way.

This could lead to the authors misinterpreting a central finding of this manuscript- that mixed dimers of unprocessed and processed (cleaved) Mpro exist in solution.

A: Based on our experience and knowledge of the technique we demonstrate that these results are both real and consistent with the rest of our data. Consequently the central finding of the manuscript holds good.

In fact, these mixed dimers could be artifacts of the dissociation/reassociation in the gas phase.

A: Thank you for the comments. Native MS is not a gas phase phenomenon. While measurement of m/z occurs in the gas phase, the assignment of multiple charges in electrospray ionisation, which allows a protein's native or denatured state to be determined, and is thus the rationale for native MS, cannot occur here. Charge assignment must occur at or prior to desolvation, which is in the liquid phase. Gas phase reassociation, which the referee mentions, is hypothetical. Because of Coulombic repulsion, it is difficult to conceptualise how two highly positively charged ions could approach each other and spontaneously associate in the gas phase. This is something we have never observed and for theoretical reasons do not believe occurs.

Additionally, the unfolded monomer observed in the native MS are consistent with unfolding within the mass spectrometer.

A: Thank you for the comments. The unfolded monomer observed is only consisted with unfolding. Where this unfolding has taken place cannot be determined. A proportion of denaturation is expected during protein purification and subsequent sample handling. The fact that the majority on the monomer remains folded is consistent with native conditions having been maintained. If significant unfolding in the mass spectrometer had occurred, the folded monomer and folded multimer structures cannot then be explained.

It should be noted that mixed tetramers were observed as the authors claim, and if such a mixed tetramer can only exist in the new model proposed here, that is evidence to support their claims. In summary the native MS is more consistent with the prevailing model than the new model proposed here. The authors note this as well "It is also clear by the mass relative quantities that the equilibria 96 of oligomeric states favor the states where more cleaved elements are present (Fig. 1 97 and S1), still befitting with that model were N-terminal cleavage is directly involved in the 98 oligomerization."

A: Thank you for your comments. The native MS shows that all combinatory possibilities of cleaved (C) and uncleaved (U) elements can occur, which is not consistent with the first model, where N-terminal must be cleaved to form dimers (therefore particles like UU, UC, UUC, UUU, UUUC and UUUU wouldn't be possible).

Another important point that the paper shows later is that the covalent inhibitor induces dimerization, rather than inhibit. If first model was correct, enzymatic activity should be directly proportional to dimerization, but an enzymatic inhibitor is strongly inducing dimerization.

The fact that more or less cleaved elements are present does not talk much about the order of events. As stated in sample prep, we arbitrarily select a sample for analysis in a given time (the purification time) that is not the equilibria, as sample is dynamic. If we measure another peak in another time, we will see different results. But assuming that equilibria will be achieved once all particles are cleaved and forming dimers (as the SECmals experiments clearly shows it happens), it is only logical that equilibria will favour states of containing more cleaved elements, if enough time is given. If we were able to purify faster, the consequence should include a different balance between C and U elements.

To prove native conditions under their unusual experimental conditions an oligomeric control such as aldehyde dehydrogenase or GROEL would be necessary.

A: Thank you for your suggestions. This is the run of E. coli beta-galactosidase LacZ, a well characterized tetramer control, showing the validity of our setup.

For other references of our group using this methodology, please see:

<https://doi.org/10.1042/BCJ20170527>

<https://doi.org/10.1074/jbc.M115.683268>

<https://doi.org/10.1038/s41467-018-04735-2>

5) The authors do not give sufficient rationale for their experimental design. Why is it important that a covalent ligand (which we must assume was intended to bind Cys145) binds C145S Mpro? Couldn't such a ligand be binding off target, and if so, where is it binding? It is unclear why this system (pertaining to the last third of the manuscript) should be taken as representative of wild-type Mpro.

A: Thank you for your comments. Considering that nirmatrelvir is an inhibitor of enzymatic activity, if enzymatic cleavage of N-terminal was the trigger for dimerization (as previous model intrinsically requires to be true), a potent inhibitor should prevent dimerization, as we saw for the non- covalent.

On the contrary, we show with our SEC mals that the covalent inhibitor induces the dimerization of samples. Further, we showed with the x-ray structure that N-terminal is not cleaved, but still particles form dimers.

So, we could only conclude that enzymatic activity was not important for dimerization, but the covalent binding was. We made minor adjustments to the form of writing to improve its understanding.

The x-ray structure of C145S with Nirmatrelvir shows is not binding off target, and the binding mode is (almost) identical to WT, see below (blue is C145S and green is WT)

This is also another confirmation for our hypothesis that the C145S model works as a slow version of Mpro.

6) Confirmation of some results with wild-type Mpro would be welcome.

A: Thank you for your comments. Native Mpro results such as crystal structure, biochemical characterization, comparative SEC mals, compared kinetic and xray structures were previously reported in our parent paper at <https://doi.org/10.1016/j.jmb.2021.167118> . The heterologous native Mpro is purified as full active dimers, and there is very little one can deduce from the maturation process by looking to the end of it. We have referenced the above mentioned manuscript where we found it suitable to show mpro WT as a control.

7) Both the native MS and cryoEM results seem to favor the prevailing model, rather than the novel model proposed here.

A: Thank you for the comments. The native MS shows that folded oligomers can be composed by combination of cleaved (C) and uncleaved (U) particles, which contradicts the previous model. If prevailing model was correct, we would not see dimers UU or UC, only CC. The same goes for trimers and tetramers.

The cryo-EM is a snapshot from a selected fraction of a complex sample that contains mixed oligomeric states, where we expect the majority of particles are forming tetramers due our arbitrary selection of taking a specific peak from SEC. This structure reveals what tetramers truly are (folded dimers carrying its substrate, aka uncleaved particles), but cryoEM cant be used to distinguish what is the composition of that sample, because we are only able of reconstruct model when particles are good enough, which does not exclude the existence of other particles classes in that sample. Moreover, in our view, the cryoEM results do not advocates for any of the two models without the context, because none of the models exclude the existence of this step during the maturation.

Minor comments:

"To obtain heterologous mature Mpro" should be reworded. Do the authors mean "heterologously expressed, mature Mpro?"

A: line 58 was modified to ` To obtain heterologous expressed mature Mpro`

"The final model showed none rotamers" Do the authors mean "nine rotomers"?

A: we mean ` no rotamers, C β or Ramachandran outliers`, meaning that general stereochemistry is of quality. No change was performed

Reviewer #2 (Remarks to the Author):

This is a potentially significant study on the oligomeric state of the SARS-COV-2 main protease Mpro in solution, and the effect of inhibitors in this process, and the possible involvement of N-terminal processing. The main finding of their study is that the key dimerization process is not triggered by N-terminal processing, but by substrate processing.

The paper is well researched and clearly described. My only real concern is the choice of techniques that were used by the Researchers, particularly as important potentially concentration dependent associative processes were being studied and conclusions being drawn thereof.

Mass spectroscopy is not a solution method and whereas SEC-MALS is a very powerful solution method for the analysis of polydisperse and heterogeneous systems that are not concentration dependent, the necessity of a separation matrix and issues of concentration control creates difficulties for concentration dependent reversible processes. How do dilutions on the columns affect the oligomerization behaviour? Only one injection concentration seems to have been attempted. I would have greater confidence in their conclusions, particularly based on Fig 5 if they had been reinforced by different injection concentrations, and more importantly also if the results had been supported by measurements from the analytical ultracentrifuge.

A: Thank you for your comments. Indeed, SEC-MALS was critical for mapping the dynamic and equilibria point of samples, which seem to converge to dimers no matter what. This method was standardized in our previous publication (<https://doi.org/10.1016/j.jmb.2021.167118>), and all measurements so far have been consistent since. The results of multiple injections containing different concentrations were added to Supplementary Figure 4, showing that different concentrations of sample 2 will still produce consistent results, as requested by the reviewer.

It is important to notice that the conversion of mixed oligomeric states into dimers by both samples has been broadly used both in this and in our previous publication, showing consistent result with other canonical methods, such as activity assay, sds page analysis, native mass and crystallography.

Plus, we have followed reviewer suggestion of checking oligomeric concentration dependency using analytical ultracentrifuge. We tested how sample 2 behaves in two distinct concentrations (1 and 2 mg/mL) and observed no significant change in relative peak patterns between these two concentrations (now added to Sup Figures). See plots bellow

A revised paper with these critical additional measurements would be more useful.

Reviewer #3 (Remarks to the Author):

SARS-CoV2 main protease is a promising drug target proven by the matured drug trials targeting this protein. Understanding the maturation process and elucidating the conformational changes associated with every steps will help to develop more powerful therapeutics. With this goal in mind, Noske et al have presented this manuscript providing a characterisation of the protein by an ensemble of techniques including Native mass spectrometry, cryo-EM and crystallography. The main points are

- i) capturing the structure of the dimeric protein bound to its uncleaved substrate. This structure, although stands against the existing model, wherein maturation of the protease by its self cleavage is crucial for its dimerisation to form active protease, it complements the missing information that helps to understand the maturation process better. Along with publications from other groups, the current author's earlier paper <https://doi.org/10.1016/j.jmb.2021.167118>, shows that any presence of additional residues in its N-terminus leads to monomeric and less active enzyme called immature MPro. In this line, the finding here sheds new insights into the maturation process of the main protease.

ii) Experimental proof of the equilibria shifting to dimeric form while bound to a covalently linked inhibitor and to monomeric while bound to a non-covalently bound inhibitor will be exploited for designing better therapeutics.

The study is well planned and sufficient experimental proofs have been given. I recommend the acceptance of the manuscript. Some important points to be looked into before publication are:

1. The cryo-EM model has some clashes, especially, between the peptide and the main protease. This needs to be addressed and corrected and accordingly the protein-peptide interaction details need to be amended in the manuscript. Also, clashes at the interface, if any, needs to be addressed.
A: Thank you for your comments. The model was updated according to reviewer requests to improve stereochemistry specially in the substrate, and reposit under the code 8EY2. Some clashes are still present, but is we found impossible to fix them without overfit our data. Please keep in mind that this is a 3.5Å cryoEM model, so zero stereochemistry outliers is very unrealistic, unless we overfit our model. We also state that our Molprobrity score puts this sample among the 2% best structures in PDB, and clashscore puts among the 14% best overall resolution, so we believe this is close to the best model possible for this map. Bellow comparative tables for old and new depositions

Old		New	
Particles	227,533	Chains	4
Chains	4	Atoms	4908
Atoms	4904	Residues	634
Residues	634	Bonds (RMSD)	
Bonds (RMSD)		Length (Å)	0.003
Length (Å)	0.005	Angles (°)	0.6
Angles (°)	0.72	MolProbrity score	2.04
MolProbrity score	2.22	Clash score	13.98
Clash score	17.6	Ramachandran plot (%)	
Ramachandran plot (%)		Outliers	0
Outliers	0	Allowed	5.75
Allowed	7.7	Favored	94.25
Favored	92.3	Rotamer outliers (%)	0
Rotamer outliers (%)	0	Cβ outliers (%)	0
Cβ outliers (%)	0	Peptide plane (%)	0
Peptide plane (%)	0	CaBLAM outliers (%)	1.46
CaBLAM outliers (%)	2.43	Box	
Box		Lengths (Å)	72.48, 86.85, 74.44
Lengths (Å)	73.79, 88.81, 73.79	Angles (°)	90, 90, 90
Angles (°)	90, 90, 90	Masked dFSC model (0/0.143/0.5)	3.3/3.5/3.8
Masked dFSC model (0/0.143/0.5)	3.3/3.5/3.8	Model vs. Data	
Model vs. Data		CC (mask)	0.80
CC (mask)	0.82	CC (box)	0.78
CC (box)	0.78	PDB code	8EY2
PDB code	7S82	EMDB	EMD-28666
EMDB	EMD-24889	EMPIAR	EMPIAR-10810
EMPIAR	EMPIAR-10810		

2. The 8dg3 model has some ramachandran outliers, which needs to be corrected as well.

A: Model was improved to fix Ramachandran outliers and redeposited under the code 8EYJ

3. It is mentioned beyond Met6 the density is not visible in the cryo-EM structure. However, it is not clear how many residues are invisible? This detail could also be included in the cloning part.

**A: Following reviewer suggestion, in the methods, we described the N-termini AA construct as `This construct contains a N-termini 6x histidine tag followed by a TEV cleavage site, and the nsp4 C-termini recognition sequence amino acids (Ser-4, Ala-3, Val-2, Leu-1, Gln-0.) followed by the Mpro sequence containing C145S substitution`
 In the results and discussion, we specified that Met6 is from Mpro, and the visible residues are from nsp4-nsp5 (residues -4 to +6). These changes were added to line 361-364**

4. The proof shown from raw low-pass filtered particles for Mpro monomeric unfolded particles (Fig 4D) seem very weak. There is a possibility that these particles could be formed during vitrification due to air-water interface. As the authors could not produce a 2D averages of these due to the mobility, statistics in the form of percentage of these particles in this data set could help.

A: Thank you for your comments. Indeed, we agree with the reviewer that the low-pass filtered images of particles is not the most ideal proof. However, in the context where we clearly see the N-terminal uncleaved portion bound to its specific position in active site (therefore knowing the satellite particles position), and native mass allowed us to confirm the presence of these particle in the sample, we cannot imagine any other possibility rather than the one presented. In our opinion, the likelihood of these particles forming a complex to be a by-product of vitrification is the same that for any other cryoEM complex with supporting biological data.

Also in our opinion, visually inspecting particles to try determining percentage and composition of oligomers is unlikely to produce any reliable information, as orientations of both central and satellite particles are random (keeping in mind that Mpro is a 60 kDa protein so signal is very small, putting our structure at the top of smallest proteins solved by this date)

The 3DVA tool in cryosparc should be able to distinguish between classes containing empty or half empty active site, but all generate 3D classes showed the similar structure, with the peptide in the active site. More recently, a new tool released in cryosparc v.4 called 3D classification was also runned in the data, given similar results (not showed in the paper). These two state of the art analysis indicate that 100% of particles (or close to it) are bound to two substrates peptides, suggesting that other particles were filtered during data processing.

We have exhaustively tried to generate 2d classes of this satellite particles wobbling around Mpro, but the nature of their movement is too random for generating consistent data. The best we could obtain were images like this, where we only see one of the satellites, likely because positioning two to signal sum is too rare.

Still, we believe that the particle and structural inspection, and the sample quality summed with native mass results, as well as the biological meaningfulness of these findings strongly point to the conclusion presented in our model. These changes were added to line 182-183

Also, there is no mentioning of how long after purification the cryo-EM was performed, for ex, right after purification indicating the sample at 0h, as to judge whether lots of these unfolded particles are anticipated.

A: Thank you for this suggestions. Since this is a very time dynamic sample, every experiment started with samples at 0h after purification. We included a more precise description of samples aliquoting and preparation in methods and clarify to each experiment which sample was used and in which time. We also added the SEC chromatogram to the Supplementary material and depicted precisely which peaks were forming each sample.

Minor remarks:

1. It should be mentioned that the surface shown in Fig3A is from the cryo-EM reconstruction or generated from the 3D model fitted in the cryo-EM map.

A: We modified the text to `Four-sides rotation view of final map displayed as surface, with chains A and B coloured in white and grey, respectively, and active site peptide map coloured in cyan`, as suggested by the reviewer at line 721

2. The manuscript should be thoroughly read for consistency in NSP or nsp notations, repetition of words, spelling errors, etc. I found many but could list only some below

A: fixed nsp notation according suggestion. Many other misspelling were fixed in this version.

3. Line 126 RMSD to 3 decimals not needed.

A: fixed according suggestion

4. Line: 170 Leucine and valine: "e" is missing.

A: fixed according suggestion

5. Fig 5D should be Fig 6C

A: fixed according suggestion

6. Line 273: should be "with the exception"

A: fixed according suggestion

7. Line 274: Gln189 "6" ?

A: fixed according suggestion

REVIEWERS' COMMENTS

Reviewer #1 (Remarks to the Author):

Previous studies indicated that formation of the mature Mpro dimer requires cleavage of the Mpro Nterminus (model 1). This manuscript proposes that Mpro dimer formation does not require dimer formation (model 2). The native mass spectrometry data, which shows mixed oligomers (processed and unprocessed N-termini), are consistent with this model. However, as detailed below, the native MS data lack the controls that could exclude artifactual formation of the observed mixed oligomers. The second major line of evidence is substantial, and shows that with a covalent inhibitor, unprocessed Mpro can form dimer. In addition, this manuscript provide valuable structural information of processing intermediates. The interpretation of the results could be improved. Two models are pitted against each other as if they are mutually exclusive, and as a result of this binary interpretation, some readers will get the impression that N-terminal processing of Mpro is not important for Mpro dimer formation (the data do not support this). All the data still support the previous model of the authors that N-terminal processing promotes dimer formation, and examples of an unprocessed dimer's existence do not exclude this model. In other words, by attacking a strawman (N-terminal processing is absolutely required) rather than considering rates (N-terminal processing speeds maturation and decreases the dimer's Kd) the authors miss an opportunity for circumspect analysis. These shortcomings notwithstanding, the revised manuscript is much improved, and has troves of structural data that will be of value to the community. With the changes outlined below, and hopefully a move towards a model that incorporates past an previous data, merits publication. Minor comments are listed below, followed by responses to the author's revisions/rebuttal (in blue) along with the first review.

Minor comments:

Please name the form of MPRO used in Fig 1 (i.e., the variant)

Keep same labeling scheme (colors) for MS data in Fig 1 and supplementary figures, when possible.

Please undertake an in depth edit on the supplementary figure legends, which lack sufficient description and have typographical errors.

Supplementary Fig 1 legend , check figure legend, it appears that label (D, E) for the last sentence is incorrect. Did you mean "E, F"

Supplementary Fig 6 legend, there's not even labels describing the difference between the top and bottom panels.

[Start previous review, reviewer's responses to revised manuscript in blue]

Reviewer #1 (Remarks to the Author): The main innovation of this manuscript is that that oligomeric states of Mpro are not dependent of N-terminal processing, which the authors claim is in opposition to prevailing expert opinion. They work with an inactive (or slow) active site variant, Cys145S, drawing conclusions from this. This manuscript was not written clearly from this reviewer's point of view, making it very difficult to review. It is possible that some of the concerns raised below could be addressed in a work that is rewritten to enumerate the rationale for the study design together with a diagram illustrating the key steps in maturation. I do not recommend publication of this manuscript in its current form.

Major concerns: 1) The literature review appears to be incomplete and provides few references, other than those of the authors, to studies and structures that support or refute the claim that the oligomeric states of Mpro are not dependent upon N-terminal processing.

A: Thank you for the comments. We included more references in the second paragraph of discussion

section. We presented similar studies that have been done using a mutated version for SARS1 Mpro and found similar results, as well as studies showing that a in-vitro model of the polyprotein still dimerizes in presence of substrate. These modifications are depicted across lines 333-345.

Second review: Thank you, this concern was addressed.

2) The authors published (2021) an X-ray study featuring the same CysSer mutant studied in this manuscript. They do not clearly enumerate the major differences and similarities between their new EM structure and old X-ray structure. They describe the EM structure in great detail, but the analysis of these details into a coherent structural framework is lacking.

A: Thank you for the comments. In the `old` xray structure, we saw the active site of chain A bound to N-terminal peptide post cleavage. In that structure, nsp4 aa Ser-4, Ala-3, Val-2, Leu1, Gln-0 were found bound to positive sites of Mpro, probably due to the slow cleavage process during the crystallization setup. In here, cryo-EM structure of freshly prepared C145S bound to N-terminal reveals details of the pre-cleavage process, which serves as a model for the initial step of the maturation process. Our model was better explained in this new version and validate with extra native ms data.

Second review: Thank you, this concern was addressed.

3) The terminology should be more precise. "Oligomeric forms" would apply to the dimer, trimer, and tetramer. The authors use "oligomeric" when only "dimeric" would justify the conclusions they draw.

A: Thank you for your suggestions. We followed the terminology proposed by the reviewer to avoid redundancy. These modifications are depicted in the abstract, and lines 295, 330-345,

Second review: Thank you, this concern was addressed.

4) The native MS analysis has serious flaws, to the extent that the conditions used were probably not native. Most native MS studies (of oligomers) are performed using nanoflow ionization (at very low flow rates and very small internal diameter tips) and higher concentration buffers (100-200 uM). This enables the use of gentle desolvation, e.g. lower capillary temperatures and lower voltages, i.e., in regions of hypersonic expansion that can dissociate oligomers and unfold proteins. The high flow, high temperature, high voltage conditions used by the authors likely result in dissociation of the tetrameric forms of Mpro to smaller trimers, dimers, and monomers (and likewise the dissociation of the trimeric form).

A: Thank you for the comments and suggestions. Using the method we describe, we observed an ion distribution for MPro monomer in its folded form which is clearly distinct from the smaller distribution of the unfolded (denatured monomer). The charge separation and intensity distribution of the larger monomer is characteristic of protein in the native condition, whereas the smaller monomer is characteristic for the denatured condition. These clear differences form the basis for interpretation of all native MS results however they are produced. The protein complexes observed with stoichiometry consistent with our other observations and predicted by the MPro activation mechanism is difficult to explain as anything other than native structures. We believe our native MS conclusions are unequivocal and self-evident. Moreover, the conditions we use maintain the native state. Reviewer number 1 is correct that most laboratories use nanoflow and the static infusion method for acquisition of native spectra. However their understanding of the ESI process is incomplete. Complete desolvation can be achieved using counter-flow drying gas at high temperature. During this desolvation, latent heat of evaporation is drawn from the electrospray droplet itself causing cooling and maintaining the temperature within droplet close to ambient, thus maintaining native conditions. The unequivocal evidence for this is that functionally and structurally correct proteins and complexes are routinely observed this way.

Second review: Thank you for the detailed response. I disagree with the statement that "However their [the leading native MS laboratories in the world] understanding of the ESI process is incomplete." I have discussed the ESI process in depth with many of these PIs, and they seem to

understand it. What the authors state about evaporation cooling of the droplet maintaining the droplet temperature to ambient is true initially, but if the analyte is desolvated half-way through a relatively hot (compared to nanospray) capillary, its temperature has to increase and it can partially unfold and quaternary structures can dissociate. The latter (dissociation) is generally more difficult to prevent than the former (unfolding). If the analyte hasn't desolvated within the capillary, larger voltages are required (compared to nano) in the nozzle-skimmer region, potentially having the same deleterious effects. This has been extensively investigated, and a good example by a well-respected scientist can be found (specifically dealing with problems associated with high flow ESI, <https://link.springer.com/article/10.1007/s13361-010-0052-1>) starting with the sentence "Conversely, protein complexes may get disrupted during or after ESI, for example by collision-induced dissociation (CID) [19, 35, 46–49]." I remain concerned that the smaller oligomers could be artifacts of the dissociation of larger in-solution oligomers, e.g., peaks attributed to trimers, dimers, and monomers (both folded and unfolded) are artifacts of dissociation of a tetramer. Moreover, when quaternary structures dissociate, charges do not partition evenly, such that a folded tetramer could dissociate into an unfolded monomer. The excellent positive control provided by LacZ somewhat assuages my concerns.

This could lead to the authors misinterpreting a central finding of this manuscript- that mixed dimers of unprocessed and processed (cleaved) Mpro exist in solution.

A: Based on our experience and knowledge of the technique we demonstrate that these results are both real and consistent with the rest of our data. Consequently the central finding of the manuscript holds good.

Second review: Thank you, see my detailed response above regarding potential pitfalls.

In fact, these mixed dimers could be artifacts of the dissociation/reassociation in the gas phase. A: Thank you for the comments.

Native MS is not a gas phase phenomenon. While measurement of m/z occurs in the gas phase, the assignment of multiple charges in electrospray ionisation, which allows a protein's native or denatured state to be determined, and is thus the rationale for native MS, cannot occur here. Charge assignment must occur at or prior to desolvation, which is in the liquid phase. Gas phase reassociation, which the referee mentions, is hypothetical. Because of Coulombic repulsion, it is difficult to conceptualise how two highly positively charged ions could approach each other and spontaneously associate in the gas phase. This is something we have never observed and for theoretical reasons do not believe occurs.

Second review: Thank you. I agree completely that association of two highly charged, desolvated proteins does not occur in the gas phase. However, I remain concerned that monomers and dimers in the solution phase can create gas phase artifacts during ionization. Again, I will defer to expert opinion, same link as above. "Of greater concern is the possibility that nonspecific complexes can be formed from ESI droplets that contain two or more protein molecules. ESI-induced artifacts of this type can give rise to false-positive results, i.e., the observation of gas-phase complexes that did not exist in solution [17, 32, 35, 45]."

Additionally, the unfolded monomer observed in the native MS are consistent with unfolding within the mass spectrometer.

A: Thank you for the comments. The unfolded monomer observed is only consistent with unfolding. Where this unfolding has taken place cannot be determined. A proportion of denaturation is expected during protein purification and subsequent sample handling. The fact that the majority of the monomer remains folded is consistent with native conditions having been maintained. If significant unfolding in the mass spectrometer had occurred, the folded monomer and folded multimer structures cannot then be explained.

Second review: Thank you. It's better to avoid absolutes here. I agree that the fact that you observe a

mixture of folded and unfolded monomer could mean either that native conditions were maintained (and there were folded and unfolded monomers in solution) or that your conditions are harsh enough to unfold a proportion of a fully folded protein preparation. I normally wouldn't interpret the observation of an admix of tetramer, trimer, dimer, and folded monomer as evidence of having obtained fully native conditions, I would interpret it conservatively. A positive control consisting of native MS of a tetrameric preparation (like those available from your SEC analysis), would settle this (if you observe only tetramer). You could also settle this by comparing the percentages of monomer, dimer, and tetramer observed in SEC with those observed in native MS. Either of those is recommended, or alternatively, changing the text to acknowledge these potential shortcomings.

It should be noted that mixed tetramers were observed as the authors claim, and if such a mixed tetramer can only exist in the new model proposed here, that is evidence to support their claims. In summary the native MS is more consistent with the prevailing model than the new model proposed here. The authors note this as well "It is also clear by the mass relative quantities that the equilibria 96 of oligomeric states favor the states where more cleaved elements are present (Fig. 1 97 and S1), still befitting with that model were N-terminal cleavage is directly involved in the 98 oligomerization."

A: Thank you for your comments. The native MS shows that all combinatory possibilities of cleaved (C) and uncleaved (U) elements can occur, which is not consistent with the first model, where N-terminal must be cleaved to form dimers (therefore particles like UU, UC, UUC, UUU, UUUC and UUUU wouldn't be possible). Another important point that the paper shows later is that the covalent inhibitor induces dimerization, rather than inhibit. If first model was correct, enzymatic activity should be directly proportional to dimerization, but an enzymatic inhibitor is strongly inducing dimerization. The fact that more or less cleaved elements are present does not talk much about the order of events. As stated in sample prep, we arbitrarily select a sample for analysis in a given time (the purification time) that is not the equilibria, as sample is dynamic. If we measure another peak in another time, we will see different results. But assuming that equilibria will be achieved once all particles are cleaved and forming dimers (as the SECmals experiments clearly shows it happens), it is only logical that equilibria will favour states of containing more cleaved elements, if enough time is given. If we were able to purify faster, the consequence should include a different balance between C and U elements. To prove native conditions under their unusual experimental conditions an oligomeric control such as aldehyde dehydrogenase or GROEL would be necessary. A: Thank you for your suggestions. This is the run of E. coli beta-galactosidase LacZ, a well characterized tetramer control, showing the validity of our setup. For other references of our group using this methodology, please see:

<https://doi.org/10.1042/BCJ20170527> <https://doi.org/10.1074/jbc.M115.683268>

<https://doi.org/10.1038/s41467-018-04735-2> 5)

Second review: Thank you. The LacZ results are excellent. We can achieve similar results with many oligomeric proteins, but I will note that we can't with certain proteins (like hemoglobin). Hopefully your Mpro preparations are well-behaved, but until you observe that a tetrameric Mpro preparation (SEC) remains a tetramer within the MS, we can't be sure (and you could have done this- the tetramer takes two days to become dimer). I agree that your results with a covalent inhibitor are very important and support your hypothesis, and in the revised manuscript that was clear.

The authors do not give sufficient rationale for their experimental design. Why is it important that a covalent ligand (which we must assume was intended to bind Cys145) binds C145S Mpro? Couldn't such a ligand be binding off target, and if so, where is it binding? It is unclear why this system (pertaining to the last third of the manuscript) should be taken as representative of wild-type Mpro.

A: Thank you for your comments. Considering that nirmatrelvir is an inhibitor of enzymatic activity, if enzymatic cleavage of N-terminal was the trigger for dimerization (as previous model intrinsically requires to be true), a potent inhibitor should prevent dimerization, as we saw for the non-covalent. On the contrary, we show with our SEC mals that the covalent inhibitor induces the dimerization of

samples. Further, we showed with the x-ray structure that N-terminal is not cleaved, but still particles form dimers. So, we could only conclude that enzymatic activity was not important for dimerization, but the covalent binding was. We made minor adjustments to the form of writing to improve its understanding. The x-ray structure of C145S with Nirmatrelvir shows is not binding off target, and the binding mode is (almost) identical to WT, see below (blue is C145S and green is WT) This is also another confirmation for our hypothesis that the C145S model works as a slow version of Mpro.

Second review: Thank you. The importance of your nirmatrelvir versus non-covalent inhibitor are now apparent. This was very important because while I have reservations about the native MS analysis, these data are beyond reproach.

6) Confirmation of some results with wild-type Mpro would be welcome.

A: Thank you for your comments. Native Mpro results such as crystal structure, biochemical characterization, comparative SEC mals, compared kinetic and xray structures were previously reported in our parent paper at <https://doi.org/10.1016/j.jmb.2021.167118> . The heterologous native Mpro is purified as full active dimers, and there is very little one can deduce from the maturation process by looking to the end of it. We have referenced the above mentioned manuscript where we found it suitable to show mpro WT as a control.

Second review: Thank you for addressing this. I note for future reference that the use of covalent inhibitors should permit the purification of nascent Mpro.

7) Both the native MS and cryoEM results seem to favor the prevailing model, rather than the novel model proposed here.

A: Thank you for the comments. The native MS shows that folded oligomers can be composed by combination of cleaved (C) and uncleaved (U) particles, which contradicts the previous model. If prevailing model was correct, we would not see dimers UU or UC, only CC. The same goes for trimers and tetramers. The cryo-EM is a snapshot from a selected fraction of a complex sample that contains mixed oligomeric states, where we expect the majority of particles are forming tetramers due our arbitrary selection of taking a specific peak from SEC. This structure reveals what tetramers truly are (folded dimers carrying its substrate, aka uncleaved particles), but cryoEM cant be used to distinguish what is the composition of that sample, because we are only able of reconstruct model when particles are good enough, which does not exclude the existence of other particles classes in that sample. Moreover, in our view, the cryoEM results do not advocates for any of the two models without the context, because none of the models exclude the existence of this step during the maturation.

Second review: Thank you for addressing this.

Reviewer #2 (Remarks to the Author):

The authors have addressed my previous concerns, and provided the concerns of the other reviewers have been addressed also, I can recommend publication

Reviewer #3 (Remarks to the Author):

All my raised concerns are addressed in the revised manuscript. Thank you for these findings.

Previous studies indicated that formation of the mature Mpro dimer requires cleavage of the Mpro N-terminus (model 1). This manuscript proposes that Mpro dimer formation does not require dimer formation (model 2). The native mass spectrometry data, which shows mixed oligomers (processed and unprocessed N-termini), are consistent with this model. However, as detailed below, the native MS data lack the controls that could exclude artifactual formation of the observed mixed oligomers. The second major line of evidence is substantial, and shows that with a covalent inhibitor, unprocessed Mpro can form dimer. In addition, this manuscript provide valuable structural information of processing intermediates. The interpretation of the results could be improved. Two models are pitted against each other as if they are mutually exclusive, and as a result of this binary interpretation, some readers will get the impression that N-terminal processing of Mpro is not important for Mpro dimer formation (the data do not support this). All the data still support the previous model of the authors that N-terminal processing promotes dimer formation, and examples of an unprocessed dimer's existence do not exclude this model. In other words, by attacking a strawman (N-terminal processing is absolutely required) rather than considering rates (N-terminal processing speeds maturation and decreases the dimer's Kd) the authors miss an opportunity for circumspect analysis. These shortcomings notwithstanding, the revised manuscript is much improved, and has troves of structural data that will be of value to the community. With the changes outlined below, and hopefully a move towards a model that incorporates past an previous data, merits publication.

A: Thank you for your comments. In our discussion, we said `Our model might also explain why Mpro studies that used other strategies for protein production that generated authentic N-terminus, such as the cases of MERS-CoV Mpro where authors opted for using thrombin at the N-termini (therefore, not allowing the protein to perform internal cleavage and complete maturation) had generated protein that behaves mostly as monomers 36. Curiously, authors have also reported that dimerization of MERS-CoV Mpro could be triggered by substrate, which also agrees with our proposed mode` - these observations are in line with our model, where being processed is not critical, but an external substrate can induce dimerization.

Still, we added the following statement to the last paragraph of the discussion, `Still, more data is required to conclude nascent nsp5 are processed and converted into mature, and if this process is dependent of processing event, the processed particle or a combination of both`

Minor comments are listed below, followed by responses to the author's revisions/rebuttal (in blue) along with the first review.

Minor comments:

Please name the form of MPRO used in Fig 1 (i.e., the variant)

A: Thank you for your comments. Fig 1 title now carries the name of Mpro C145S

Keep same labeling scheme (colors) for MS data in Fig 1 and supplementary figures, when possible.

A: Thank you for your comments. The color scheme was maintained

Please undertake an in depth edit on the supplementary figure legends, which lack sufficient description and have typographical errors.

A: Thank you for your comments. Many corrections were made on SMI

Supplementary Fig 1 legend , check figure legend, it appears that label (D, E) for the last sentence is incorrect. Did you mean "E, F"

A: Thank you for your comments. Correction made as requested

Supplementary Fig 6 legend, there's not even labels describing the difference between the top and bottom panels.

A: Thank you for your observation. Description of top and bottom was added to Fig S6.

[Start previous review, reviewer's responses to revised manuscript in blue – and now in green]

[Start previous review, reviewer's responses to revised manuscript in blue]

Reviewer #1 (Remarks to the Author): The main innovation of this manuscript is that that oligomeric states of Mpro are not dependent of N-terminal processing, which the authors claim is in opposition to prevailing expert opinion. They work with an inactive (or slow) active site variant, Cys145S, drawing conclusions from this. This manuscript was not written clearly from this reviewer's point of view, making it very difficult to review. It is possible that some of the concerns raised below could be addressed in a work that is rewritten to enumerate the rationale for the study design together with a diagram illustrating the key steps in maturation. I do not recommend publication of this manuscript in its current form.

Major concerns: 1) The literature review appears to be incomplete and provides few references, other than those of the authors, to studies and structures that support or refute the claim that the oligomeric states of Mpro are not dependent upon N-terminal processing.

A: Thank you for the comments. We included more references in the second paragraph of discussion section. We presented similar studies that have been done using a mutated version for SARS1 Mpro and found similar results, as well as studies showing that a in-vitro model of the polyprotein still dimerizes in presence of substrate. These modifications are depicted across lines 333-345.

Second review: Thank you, this concern was addressed.

2) The authors published (2021) an X-ray study featuring the same CysSer mutant studied in this manuscript. They do not clearly enumerate the major differences and similarities between their new EM structure and old X-ray structure. They describe the EM structure in great detail, but the analysis of these details into a coherent structural framework is lacking.

A: Thank you for the comments. In the `old` xray structure, we saw the active site of chain A bound to N-terminal peptide post cleavage. In that structure, nsp4 aa Ser-4, Ala-3, Val-2, Leu1, Gln-0 were found bound to positive sites of Mpro, probably due to the slow cleavage process during the crystallization setup. In here, cryo-EM structure of freshly prepared C145S bound to N-terminal reveals details of the pre-cleavage process, which serves as a model for the initial step of the maturation process. Our model was better explained in this new version and validate with extra native ms data.

Second review: Thank you, this concern was addressed.

3) The terminology should be more precise. "Oligomeric forms" would apply to the dimer, trimer, and tetramer. The authors use "oligomeric" when only "dimeric" would justify the conclusions they draw.

A: Thank you for your suggestions. We followed the terminology proposed by the reviewer to avoid redundancy. These modifications are depicted in the abstract, and lines 295, 330-345,

Second review: Thank you, this concern was addressed.

4) The native MS analysis has serious flaws, to the extent that the conditions used were probably not

native. Most native MS studies (of oligomers) are performed using nanoflow ionization (at very low flow rates and very small internal diameter tips) and higher concentration buffers (100-200 μ M). This enables the use of gentle desolvation, e.g. lower capillary temperatures and lower voltages, i.e., in regions of hypersonic expansion that can dissociate oligomers and unfold proteins. The high flow, high temperature, high voltage conditions used by the authors likely result in dissociation of the tetrameric forms of Mpro to smaller trimers, dimers, and monomers (and likewise the dissociation of the trimeric form).

A: Thank you for the comments and suggestions. Using the method we describe, we observed an ion distribution for MPro monomer in its folded form which is clearly distinct from the smaller distribution of the unfolded (denatured monomer). The charge separation and intensity distribution of the larger monomer is characteristic of protein in the native condition, whereas the smaller monomer is characteristic for the denatured condition. These clear differences form the basis for interpretation of all native MS results however they are produced. The protein complexes observed

with stoichiometry consistent with our other observations and predicted by the MPro activation mechanism is difficult to explain as anything other than native structures. We believe our native MS conclusions are unequivocal and self-evident. Moreover, the conditions we use maintain the native state. Reviewer number 1 is correct that most laboratories use nanoflow and the static infusion method for acquisition of native spectra. However their understanding of the ESI process is incomplete. Complete desolvation can be achieved using counter-flow drying gas at high temperature. During this desolvation, latent heat of evaporation is drawn from the electrospray droplet itself causing cooling and maintaining the temperature within droplet close to ambient, thus maintaining native conditions. The unequivocal evidence for this is that functionally and structurally correct proteins and complexes are routinely observed this way.

Second review: Thank you for the detailed response. I disagree with the statement that “However their [the leading native MS laboratories in the world] understanding of the ESI process is incomplete.” I have discussed the ESI process in depth with many of these PIs, and they seem to understand it. What the authors state about evaporation cooling of the droplet maintaining the droplet temperature to ambient is true initially, but if the analyte is desolvated half-way through a relatively hot (compared to nanospray) capillary, its temperature has to increase and it can partially unfold and quaternary structures can dissociate. The latter (dissociation) is generally more difficult to prevent than the former (unfolding). If the analyte hasn’t desolvated within the capillary, larger voltages are required (compared to nano) in the nozzle-skimmer region, potentially having the same deleterious effects. This has been extensively investigated, and a good example by a well-respected scientist can be found (specifically dealing with problems associated with high flow ESI, <https://link.springer.com/article/10.1007/s13361-010-0052-1>) starting with the sentence “Conversely, protein complexes may get disrupted during or after ESI, for example by collision-induced dissociation (CID) [19, 35, 46–49].” I remain concerned that the smaller oligomers could be artifacts of the dissociation of larger in-solution oligomers, e.g., peaks attributed to trimers, dimers, and monomers (both folded and unfolded) are artifacts of dissociation of a tetramer. Moreover, when quaternary structures dissociate, charges do not partition evenly, such that a folded tetramer could dissociate into an unfolded monomer. The excellent positive control provided by LacZ somewhat assuages my concerns.

A: Thank you for your comments. Hope we have a chance to discuss in the future

This could lead to the authors misinterpreting a central finding of this manuscript- that mixed dimers of unprocessed and processed (cleaved) Mpro exist in solution.

A: Based on our experience and knowledge of the technique we demonstrate that these results are both real and consistent with the rest of our data. Consequently the central finding of the manuscript holds good.

Second review: Thank you, see my detailed response above regarding potential pitfalls.

In fact, these mixed dimers could be artifacts of the dissociation/reassociation in the gas phase. A: Thank you for the comments.

Native MS is not a gas phase phenomenon. While measurement of m/z occurs in the gas phase, the assignment of multiple charges in electrospray ionisation, which allows a protein’s native or

denatured state to be determined, and is thus the rationale for native MS, cannot occur here. Charge assignment must occur at or prior to desolvation, which is in the liquid phase. Gas phase reassociation, which the referee mentions, is hypothetical. Because of Coulombic repulsion, it is difficult to conceptualise how two highly positively charged ions could approach each other and spontaneously associate in the gas phase. This is something we have never observed and for theoretical reasons do not believe occurs.

Second review: Thank you. I agree completely that association of two highly charged, desolvated proteins does not occur in the gas phase. However, I remain concerned that monomers and dimers in the solution phase can create gas phase artifacts during ionization. Again, I will defer to expert opinion, same link as above. "Of greater concern is the possibility that nonspecific complexes can be formed from ESI droplets that contain two or more protein molecules. ESI-induced artifacts of this type can give rise to false-positive results, i.e., the observation of gas-phase complexes that did not exist in solution [17, 32, 35, 45]."

Additionally, the unfolded monomer observed in the native MS are consistent with unfolding within the mass spectrometer.

A: Thank you for the comments. The unfolded monomer observed is only consisted with unfolding. Where this unfolding has taken place cannot be determined. A proportion of denaturation is expected during protein purification and subsequent sample handling. The fact that the majority on the monomer remains folded is consistent with native conditions having been maintained. If significant unfolding in the mass spectrometer had occurred, the folded monomer and folded multimer structures cannot then be explained.

Second review: Thank you. It's better to avoid absolutes here. I agree that the fact that you observe a mixture of folded and unfolded monomer could mean either that native conditions were maintained (and there were folded and unfolded monomers in solution) or that your conditions are harsh enough to unfold a proportion of a fully folded protein preparation. I normally wouldn't interpret the observation of an admix of tetramer, trimer, dimer, and folded monomer as evidence of having obtained fully native conditions, I would interpret it conservatively.

A: Thank you for your valuable comment on `I normally wouldn't interpret the observation of an admix of tetramer, trimer, dimer, and folded monomer as evidence of having obtained fully native conditions` - neither do we. But this is an exceptional case where the mechanism and oligomeric states are linked, and the substrate inducing dimerization is the enzyme itself.

A positive control consisting of native MS of a tetrameric preparation (like those available from your SEC analysis), would settle this (if you observe only tetramer). You could also settle this by comparing the percentages of monomer, dimer, and tetramer observed in SEC with those observed in native MS. Either of those is recommended, or alternatively, changing the text to acknowledge these potential shortcomings.

A: Thank you for your valuable comment again. We did our best to try to have a full tetrameric sample, by being peek on selecting peaks at SEC during protein prep, but there is a limit to what we can do. The sample is too dynamic, and even keeping everything cold it still dissociates into other oligomeric states. Again, we do not believe difference in ratios can influence any of the conclusions; these ratios are dependent on the time of measurement, and will shift every minute. Our description is much more qualitative.

It should be noted that mixed tetramers were observed as the authors claim, and if such a mixed tetramer can only exist in the new model proposed here, that is evidence to support their claims. In summary the native MS is more consistent with the prevailing model than the new model proposed here. The authors note this as well “It is also clear by the mass relative quantities that the equilibria 96 of oligomeric states favor the states where more cleaved elements are present (Fig. 1 97 and S1), still befitting with that model were N-terminal cleavage is directly involved in the 98 oligomerization.”

A: Thank you for your comments. The native MS shows that all combinatory possibilities of cleaved (C) and uncleaved (U) elements can occur, which is not consistent with the first model, where N-terminal must be cleaved to form dimers (therefore particles like UU, UC, UUC, UUU, UUUC and UUUU wouldn't be possible). Another important point that the paper shows later is that the covalent

inhibitor induces dimerization, rather than inhibit. If first model was correct, enzymatic activity should be directly proportional to dimerization, but an enzymatic inhibitor is strongly inducing dimerization. The fact that more or less cleaved elements are present does not talk much about the order of events. As stated in sample prep, we arbitrarily select a sample for analysis in a given time (the purification time) that is not the equilibria, as sample is dynamic. If we measure another peak in another time, we will see different results. But assuming that equilibria will be achieved once all particles are cleaved and forming dimers (as the SECmals experiments clearly shows it happens), it is only logical that equilibria will favour states of containing more cleaved elements, if enough time is given. If we were able to purify faster, the consequence should include a different balance between C and U elements. To prove native conditions under their unusual experimental conditions an oligomeric control such as aldehyde dehydrogenase or GROEL would be necessary. A: Thank you for your suggestions. This is the run of E. coli beta-galactosidase LacZ, a well characterized tetramer control, showing the validity of our setup. For other references of our group using this methodology, please see: <https://doi.org/10.1042/BCJ20170527> <https://doi.org/10.1074/jbc.M115.683268> <https://doi.org/10.1038/s41467-018-04735-2> 5)

Second review: Thank you. The LacZ results are excellent. We can achieve similar results with many oligomeric proteins, but I will note that we can't with certain proteins (like hemoglobin). Hopefully your Mpro preparations are well-behaved, but until you observe that a tetrameric Mpro preparation (SEC) remains a tetramer within the MS, we can't be sure (and you could have done this- the tetramer takes two days to become dimer). I agree that your results with a covalent inhibitor are very important and support your hypothesis, and in the revised manuscript that was clear.

The authors do not give sufficient rationale for their experimental design. Why is it important that a covalent ligand (which we must assume was intended to bind Cys145) binds C145S Mpro? Couldn't such a ligand be binding off target, and if so, where is it binding? It is unclear why this system (pertaining to the last third of the manuscript) should be taken as representative of wild-type Mpro.

A: Thank you for your comments. Considering that nirmatrelvir is an inhibitor of enzymatic activity, if enzymatic cleavage of N-terminal was the trigger for dimerization (as previous model intrinsically requires to be true), a potent inhibitor should prevent dimerization, as we saw for the non-covalent. On the contrary, we show with our SEC mals that the covalent inhibitor induces the dimerization of samples. Further, we showed with the x-ray structure that N-terminal is not cleaved, but still particles form dimers. So, we could only conclude that enzymatic activity was not important for dimerization, but the covalent binding was. We made minor adjustments to the form of writing to improve its understanding. The x-ray structure of C145S with Nirmatrelvir shows is not binding off target, and the binding mode is (almost) identical to WT, see below (blue is C145S and green is WT) This is also another confirmation for our hypothesis that the C145S model works as a slow version of Mpro.

Second review: Thank you. The importance of your nirmatrelvir versus non-covalent inhibitor are now apparent. This was very important because while I have reservations about the native MS analysis, these data are beyond reproach.

6) Confirmation of some results with wild-type Mpro would be welcome.

A: Thank you for your comments. Native Mpro results such as crystal structure, biochemical characterization, comparative SEC mals, compared kinetic and xray structures were previously

reported in our parent paper at <https://doi.org/10.1016/j.jmb.2021.167118> . The heterologous native Mpro is purified as full active dimers, and there is very little one can deduce from the maturation process by looking to the end of it. We have referenced the above mentioned manuscript where we found it suitable to show mpro WT as a control.

Second review: Thank you for addressing this. I note for future reference that the use of covalent inhibitors should permit the purification of nascent Mpro.

7) Both the native MS and cryoEM results seem to favor the prevailing model, rather than the novel model proposed here.

A: Thank you for the comments. The native MS shows that folded oligomers can be composed by combination of cleaved (C) and uncleaved (U) particles, which contradicts the previous model. If prevailing model was correct, we would not see dimers UU or UC, only CC. The same goes for trimers and tetramers. The cryo-EM is a snapshot from a selected fraction of a complex sample that contains mixed oligomeric states, where we expect the majority of particles are forming tetramers due our arbitrary selection of taking a specific peak from SEC. This structure reveals what tetramers truly are (folded dimers carrying its substrate, aka uncleaved particles), but cryoEM cant be used to distinguish what is the composition of that sample, because we are only able of reconstruct model when particles are good enough, which does not exclude the existence of other particles classes in that sample. Moreover, in our view, the cryoEM results do not advocates for any of the two models without the context, because none of the models exclude the existence of this step during the maturation.

Second review: Thank you for addressing this.